# Understanding Transformer-Based Vision Models via Modular Feature Inversion

## Abstract

Understanding the internal mechanisms of deep neural networks remains a central challenge in machine learning. In computer vision, one promising yet only preliminarily explored approach is feature inversion via inverse networks, which reconstructs images from intermediate representations using trained inverse networks. In this study, we revisit feature inversion via inverse networks, introducing a novel, modular variant that enables a computationally more efficient application of the technique while, in some cases, producing semantically more coherent image reconstructions. We apply our method to large-scale transformer-based vision models, specifically Detection Transformer, Vision Transformer, Swin Transformer, and Data-Efficient Image Transformer, analyzing the resulting reconstructions across network depth. Our main analysis compares Detection Transformer and Vision Transformer, which exhibit the most informative differences among the evaluated architectures. At the same time, results for Swin Transformer and Data-Efficient Image Transformer support the broader applicability of our framework. Our findings reveal gradual representational changes across transformer layers as a shared characteristic of Detection Transformer and Vision Transformer, as well as systematic differences in their preservation of contextual shape and fine-grained image details and their robustness to color perturbations. These findings contribute to a deeper understanding of transformer-based vision models and demonstrate the utility of modular feature inversion as an interpretability tool.

## 1 Introduction

In recent years, computer vision has increasingly shifted from convolutional neural networks (CNNs) to transformer-based vision models (TVMs) (Dosovitskiy et al., 2020; Li et al., 2023; Carion et al., 2020; Zhu et al., 2021; Zhang et al., 2022). Despite their impressive performance, the internal mechanisms that enable these models to solve complex tasks remain largely opaque, limiting our ability to interpret and understand how predictions arise (Zhang & Zhu, 2018; Fan et al., 2021; Li et al., 2022). Improving network interpretability is therefore essential for ensuring reliability, optimizing performance, and identifying potential failure modes.

Feature inversion via inverse networks, introduced by Dosovitskiy & Brox (2016), is a promising, early technique for network interpretability in deep neural networks (DNNs) for vision. Building on prior work on visualizing intermediate representations (Erhan et al., 2009; Zeiler & Fergus, 2014; Mahendran & Vedaldi, 2014; Springenberg et al., 2015), their method trains inverse networks to reconstruct input images from intermediate activations of models such as AlexNet (Krizhevsky et al., 2012). By analyzing reconstructed images across layers, this approach provides insights into the information encoded at different stages of processing.

While feature inversion via inverse networks was successfully applied to AlexNet, it has not been widely adopted for modern vision DNNs. We attribute this limited use to two main factors. Firstly, training individual inverse networks for each layer of a DNN is computationally demanding, particularly for modern large CNNs and TVMs. Secondly, the potential of using feature inversion via inverse networks for analyzing

and interpreting neural networks was only preliminarily explored by Dosovitskiy & Brox (2016), leaving much of the capabilities of the method underutilized.

In this work, we revisit feature inversion via inverse networks and propose a modular framework that decomposes DNNs into constituent modules. For each forward module, we independently train a corresponding inverse module that mirrors its information flow. We show that this design enables computationally efficient inversion of large architectures, in some cases, yielding more semantically coherent image reconstructions. We apply our method to four prominent TVMs: Detection Transformer (DETR) (Carion et al., 2020) and Vision Transformer (ViT) (Dosovitskiy et al., 2020), two pioneering architectures, as well as Swin Transformer (Swin) (Liu et al., 2021) and Data-Efficient Image Transformer III (DeiT III) (Touvron et al., 2022), which represent a hierarchical variant of ViT and a more data-efficiently trained variant, respectively. We analyze the resulting reconstructions, including those obtained from targeted image manipulations (see Figure 1 for an illustration), and use these analyses to derive and test hypotheses about the internal mechanisms of the TVMs. We present our results as a comparison of DETR and ViT, which exhibit the most pronounced and informative differences among the evaluated TVMs, and show results on Swin and DeiT III in the appendix, or use them to support hypotheses and the general applicability of our modular inversion framework.

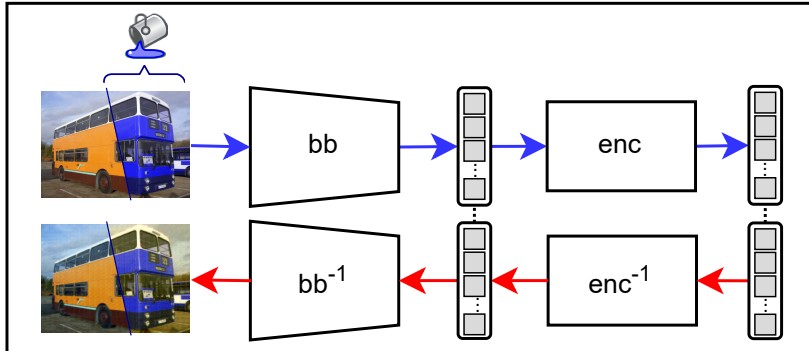 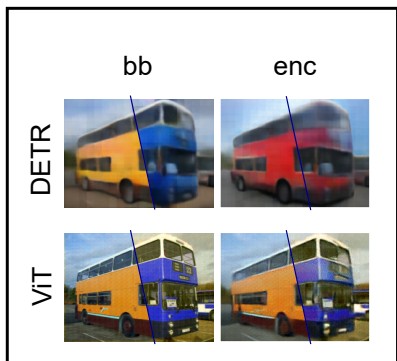

Figure 1: Illustration of our approach. **Left**: We invert modules of transformer-based vision models, such as the backbone (bb) and encoder (enc). **Right**: Using these inverted modules, we reconstruct images from different processing stages to analyze DETR and ViT. Here, we recolor a scene and observe that DETR abstracts color into prototypical representations in its encoder, whereas ViT preserves color fidelity throughout its architecture.

Our results reveal gradual changes in representations across transformer layers as a shared characteristic between DETR and ViT, alongside systematic differences in their encoding of contextual shape and fine-grained detail and in their robustness to color perturbations. Notably, despite their architectural similarities, the models exhibit distinct abstraction strategies: DETR progressively transforms object representations toward more prototypical shapes at higher layers, whereas the ViT preserves fine-grained visual detail throughout depth. We summarize our core contributions as follows:

- We introduce a novel, efficient feature inversion method based on modular, independently trained inverse modules.

- We demonstrate how reconstructed images can be systematically used to interpret internal processing mechanisms, introducing new analysis techniques such as targeted embedding manipulation.

- We identify gradual representational changes across transformer layers as a shared characteristic of DETR and ViT.

- We reveal systematic differences in image processing between DETR and ViT, particularly in terms of image detail preservation, abstraction, and robustness to color perturbations.

## 2 Related Work

In computer vision, feature inversion is a network interpretability method that reconstructs an input image from its intermediate representations of a DNN, enabling systematic inspection of the information preserved, discarded, or transformed at each processing stage.

A seminal work by Mahendran & Vedaldi (2014) formulated feature inversion as an optimization problem in image space, generating images whose intermediate representations match those of a target layer. While effective, this approach is computationally expensive and sensitive to regularization choices. Dosovitskiy & Brox (2016) addressed these limitations by introducing learned inverse networks, training dedicated models to reconstruct images from intermediate representations. Applied to AlexNet (Krizhevsky et al., 2012), this method demonstrated that both color and spatial information are preserved across layers. However, the method required training a separate inverse model for each layer, limiting scalability to modern large architectures. Moreover, its use as a systematic interpretability tool has remained largely unexplored, as its authors primarily focused on comparing inversion strategies rather than fully leveraging feature inversion via inverse networks as a tool for network interpretability.

Feature inversion is closely related to, but distinct from, activation maximization (Erhan et al., 2009; Zeiler & Fergus, 2014; Springenberg et al., 2015; Nguyen et al., 2016; Olah et al., 2017), which synthesizes inputs that maximize activations of specific neurons, channels, or layers. While activation maximization reveals the prototypical features preferred by network units, feature inversion enables investigation of the specific information retained from a given input.

More broadly, a variety of interpretability methods have been proposed to analyze vision DNNs. Attribution techniques, including saliency maps (Simonyan et al., 2014), class activation mapping (Selvaraju et al., 2017), and layer-wise relevance propagation (Bach et al., 2015), identify input regions that influence model predictions. Perturbation-based approaches, such as occlusion sensitivity (Zeiler & Fergus, 2014) and adversarial attacks (Goodfellow et al., 2015), probe input sensitivity and robustness. Representation similarity methods, including singular vector canonical correlation analysis (Raghu et al., 2017) and centered kernel alignment (Kornblith et al., 2019), quantify alignment across layers or models. Loss landscape analyses (Li et al., 2018; Garipov et al., 2018; Keskar et al., 2017) characterize the geometry of the optimization surface to study generalization and learning dynamics. Taken together, these approaches enable valuable insights into vision DNNs, but unlike feature inversion, do not provide direct, human-interpretable access to the visual content encoded in intermediate representations.

Interpretability studies of TVMs, particularly ViT, have primarily relied on the aforementioned broader techniques rather than feature inversion. These include attention rollout (Abnar & Zuidema, 2020), relevance propagation (Chefer et al., 2021), robustness and sensitivity studies (Naseer et al., 2021), and representational similarity analysis (Raghu et al., 2021). Collectively, these works suggest that ViT preserves spatial information and gradually refines representations across layers, while exhibiting more uniform representations across depth than CNNs.

In contrast to ViT, interpretability studies on other TVMs like DETR, Swin, or DeiT III remain limited (Chefer et al., 2021). Instead of analyzing existing models, recent works, particularly for DETR, have focused on architectural changes to improve interpretability by design, e.g., by incorporating feature disentanglement techniques or introducing new modules designed to learn prototypical features, thereby making subsequent interpretation more tractable (Yu et al., 2024; Paul et al., 2024; Rath-Manakidis et al., 2024).

To the best of our knowledge, feature inversion has not been systematically applied to TVMs for interpretability. Inverting self-attention and cross-attention layers is particularly challenging, as these mechanisms dynamically aggregate information across tokens, entangling spatial and semantic features in a non-local manner (Fantozzi & Naldi, 2024; Bibal et al., 2022), which impedes classical monolithic inversion approaches. Our work addresses this gap by introducing modular feature inversion as a scalable, semantically grounded tool to examine the interpretability of large-scale TVMs.

# 3 Methods

## 3.1 Feature Inversion

Feature inversion attempts to reconstruct input images from their intermediate representations within a neural network, enabling analysis of the information encoded at different processing stages. To formalize the method, let $\mathcal{N} : \mathcal{X}_0 \to \mathcal{X}_L$ be a neural network with parameters $\theta$, mapping from an image space $\mathcal{X}_0$ to an output space $\mathcal{X}_L$. We consider a set of processing stages of interest indexed by $\mathcal{P} := (1, \ldots, n)$ corresponding to selected layers of the network. Given an input image $\mathbf{x}_0 \in \mathcal{X}_0$, we denote its representation at stage $j \in \mathcal{P}$ as $\mathbf{x}_j := \mathcal{N}_{0:j}(\mathbf{x}_0; \theta_{0:j})$, where $\mathcal{N}_{i:j} : \mathcal{X}_i \to \mathcal{X}_j$ denotes the subnetwork from layer $i$ to $j$.

Furthermore, we define the approximate inverse of the subnetwork $\mathcal{N}_{i:j}$ as the neural network $\mathcal{N}_{j:i}^{-1} : \mathcal{X}_j \to \mathcal{X}_i$ with parameters $\phi_{j:i}$. The reconstruction of $\mathbf{x}_i$ from processing stage $j \in \mathcal{P}$ is given by $\hat{\mathbf{x}}_{j:i} := \mathcal{N}_{j:i}^{-1}(\mathbf{x}_j)$. When $i = 0$, we refer to the reconstruction as image reconstruction and layer reconstruction otherwise.

Feature inversion via inverse networks by Dosovitskiy & Brox (2016), which we will refer to as classical feature inversion, follows two steps. First, separate inverse networks $\mathcal{N}_{j:0}^{-1}$ are trained for each $j \in \mathcal{P}$ by minimizing the expected mean squared error (MSE) between image reconstructions $\hat{\mathbf{x}}_{j:0}$ and their corresponding input images $\mathbf{x}_0$. Then, the inverse networks are used to generate reconstructed images from the various processing stages. If the inverse networks are sufficiently powerful, the reconstructed images reflect the abstractions and omissions of features inherent to the representations $\mathbf{x}_j$ at the pixel level, enabling the assessment of what information is retained, omitted, or abstracted at different processing stages.

## 3.2 Modular Feature Inversion

Instead of training inverse networks to map directly from $\mathcal{X}_j$ to $\mathcal{X}_0$, we propose training local inverse modules between consecutive stages. Specifically, for each $j \in \mathcal{P}$, we train an inverse module $\mathcal{N}_{j:j-1}^{-1} : \mathcal{X}_j \to \mathcal{X}_{j-1}$. Each inverse modules is trained by minimizing the expected MSE:

$$L_{\text{MSE}}(\phi_{j:j-1}) := \mathbb{E}\left[\|\mathbf{x}_{j-1} - \mathcal{N}_{j:j-1}^{-1}(\mathbf{x}_j)\|_2^2\right]. \tag{1}$$

With the modular approach, we then obtain image reconstructions by sequentially applying the trained inverse modules from any processing stage $j \in \mathcal{P}$:

$$\hat{\mathbf{x}}_{0:j} := (\mathcal{N}_{1:0}^{-1} \circ \cdots \circ \mathcal{N}_{j:j-1}^{-1})(\mathbf{x}_j) \tag{2}$$

Our modular approach offers several advantages. Firstly, the inverse mapping structurally mirrors the forward pass by enforcing alignment at intermediate representations, whereas classical end-to-end inversion may deviate substantially from the forward computation. This correspondence suggests that reconstructions more faithfully reflect the transformations performed by the network. Secondly, computational efficiency is greatly improved, as fewer, smaller modules are required compared to training larger separate inverse networks for each processing stage.

This efficiency can be illustrated by comparing the total number of trainable parameters in the classic approach of feature inversion versus our modular approach: Let $\mathcal{N}$ be a DNN with $p$ parameters and $n$ processing stages of interest, for simplicity, each with $p/n$ parameters. In the full-path approach, $n$ inverse networks are trained, each inverting an increasingly larger network portion. Assuming each inverse network roughly mirrors its forward path, the total parameter count for all inverse networks is $\sum_{i=1}^{n} i \cdot \frac{p}{n} = \frac{np+p}{2}$, scaling linearly with $n$. In contrast, for the same $\mathcal{N}$, our modular method uses $n$ inverse modules of size $p/n$, totaling $p$ parameters, constant in $p$ and independent of $n$.

## 3.3 Application to Transformer-Based Vision Models

We applied modular feature inversion to pretrained base variants of DETR, ViT, Swin, and DeiT III (DETR-R50, ViT-B/16, Swin-B, DeiT-B), which provide a reasonable compromise between performance and size

(see Figure 2 for an illustration of our approach on DETR; the same procedure is applied analogously to the other TVMs).

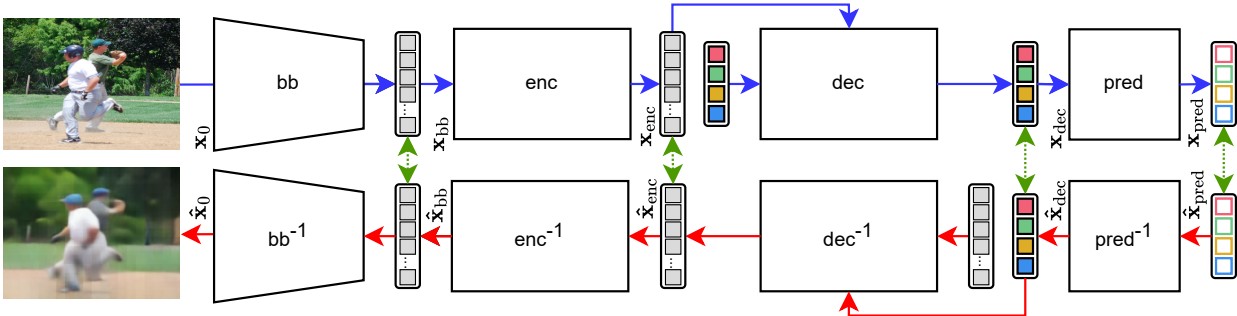

Figure 2: Modular feature inversion of DETR. Illustration of the main modules of DETR (blue), and our modular inversion approach (red). Green double-arrows indicate correspondence between representations, i.e., the outputs of a forward module can be used as inputs to its corresponding inverse module during inference, and the inputs of a forward module can be used as a supervision signal during training.

For most inverse modules, we followed the practice from classical feature inversion, i.e., inverse modules were designed to approximately mirror their corresponding forward modules.

For DETR, we identified four processing stages of interest corresponding to the outputs of its backbone (bb), its encoder (enc), its decoder (dec), and its prediction head (pred). We adopt a simplified notation, writing bb for $\mathcal{N}_{0:1}$, bb$^{-1}$ for $\mathcal{N}_{1:0}^{-1}$, and $\mathbf{x}_{\mathrm{bb}}$ for $\mathbf{x}_1$, with analogous notation for other modules (e.g., enc for $\mathcal{N}_{1:2}$). Reconstructions from a specific stage are denoted accordingly, e.g., $\hat{\mathbf{x}}_{\mathrm{enc:0}}$ represents an image reconstructed from the encoder stage.

We implemented a deconvolutional network as bb$^{-1}$, loosely resembling an inversion of its backbone (ResNet-50 (He et al., 2016)). Notably, in contrast to the other inverse modules, the parameter count of bb$^{-1}$ for DETR substantially exceeded that of its forward counterpart, as this configuration yielded significantly improved reconstruction performance, an effect not observed for any other module. We set enc$^{-1}$ to be structurally equivalent to enc. Similarly, we defined dec$^{-1}$ as structurally equivalent to dec, but initialized its input as blank tokens that self-attend to each other and cross-attend to $\mathbf{x}_{\mathrm{dec}}$. For pred$^{-1}$, we used a simple multilayer perceptron (MLP) that takes the concatenation of the bounding box and the full distribution of class logits as input.

For ViT, we considered two stages corresponding to the initial linear patch embedding, denoted as bb and encoder (enc) output. We employed a local small deconvolutional network as bb$^{-1}$. Although bb in ViT is a simple, invertible linear transformation, its analytical inverse proved to be ill-conditioned, making it highly sensitive to layer reconstruction errors between $\mathbf{x}_{\mathrm{bb}}$ and $\hat{\mathbf{x}}_{\mathrm{enc:bb}}$. For enc$^{-1}$, we used a structurally equivalent module to enc. We note that the ViT bb is substantially smaller than the ResNet backbone in DETR and may therefore differ from a standard conception of a backbone. Nevertheless, we use the term backbone consistently here to distinguish transformer-stage processing from the preceding processing steps.

For inverting Swin, we identified five processing stages of interest: the initial patch partitioning step (also denoted as bb for consistency) and the outputs of the four hierarchical stages prior to each patch merging operation ($s_1, s_2, s_3, s_4$). For bb$^{-1}$, we used the same architecture as for ViT. To invert the Swin Transformer blocks in the hierarchical stages, we employed architecturally equivalent blocks, but replaced the downsampling operations with upsampling operations, reflecting the reversed information flow in which window and patch resolutions increase rather than decrease.

We identified the same processing stages of interest for DeiT III as for ViT and employed architecturally equivalent inverse modules, given that DeiT III shares the same underlying architecture as ViT. The primary difference between the two models lies in their training procedures: DeiT III relies on a distinct data augmen-

tation strategy, enabling it to achieve comparable performance to ViT, which is trained on a substantially larger dataset with comparatively minimal augmentation.

We trained multiple instances of each inverse module, exploring different hyperparameters and, in the case of $bb^{-1}$ for DETR, slightly different architectures (see Appendix B, particularly, Appendix B.1, and the accompanying code repository[1] for details). For our analysis, we selected one fully trained instance per inverse module based on MSE performance. We generated all results reported in Section 4 with these selected instances and used the remaining modules for sanity checks to ensure that our findings were not attributable to random variation.

## 4 Results

### 4.1 Preliminary Analysis

To build intuition for our modular feature inversion approach, we begin by visualizing image reconstructions from various processing stages in DETR and ViT (see Figure 3). Early-stage image reconstructions from both models maintain the overall scene layout and coarse object structure. However, ViT preserves fine-grained details more accurately, while DETR reconstructions show signs of blurring already at the backbone stage. These differences become more pronounced in deeper stages, where DETR reconstructions progressively lose structural detail, introduce color shifts, and abstract away background elements, suggesting a systematic abstraction process. In contrast, ViT reconstructions remain comparatively faithful across stages.

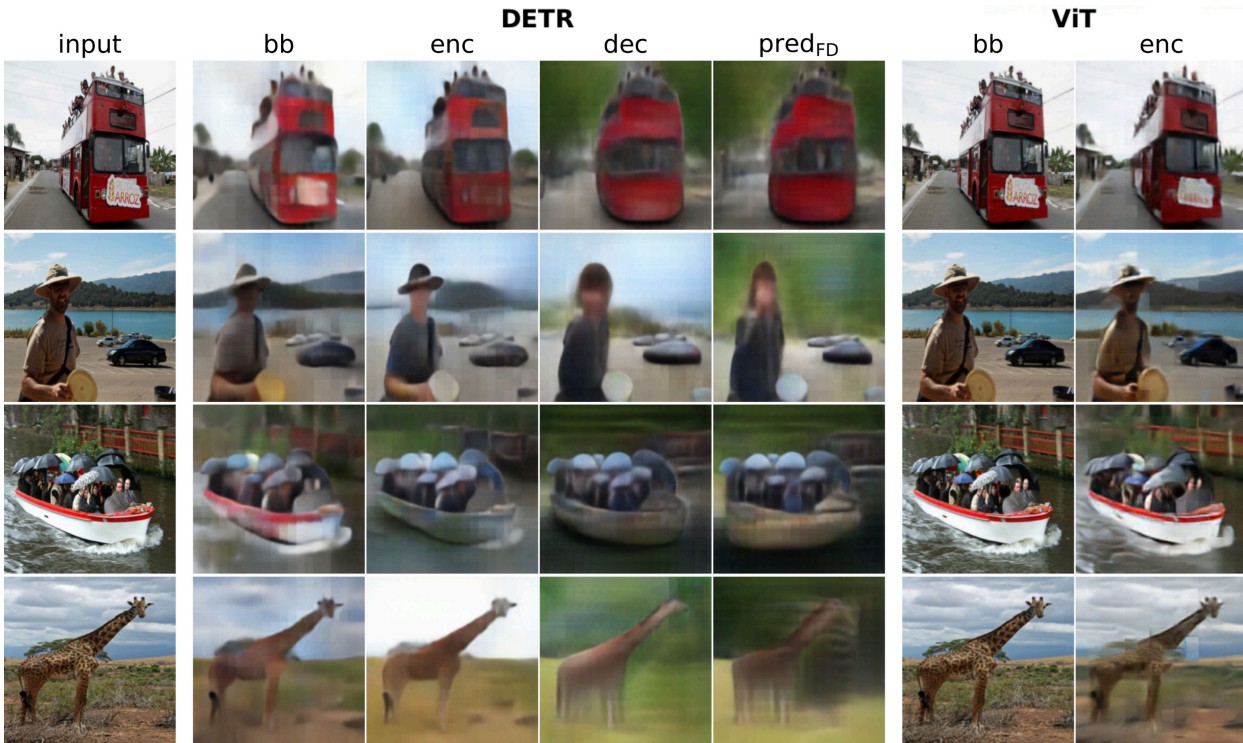

Figure 3: Image reconstructions from processing stages. Column 1 shows the original input images. Columns 2-5 and 6-7 show reconstructions from different processing stages of DETR and ViT, respectively.

As an initial quantitative assessment, we quantify the observed visual differences using the average MSE across processing stages in Figure 4 (left). As expected, reconstruction error increases at later stages in both models, with ViT maintaining significantly lower MSE than DETR throughout. Interestingly, in DETR, the MSE from the decoder stage onward exceeds the baseline error of comparing each image to the dataset

---

[1]Code submitted as supplementary material to ensure anonymity during double-blind peer review.

mean (a grayish, structureless reference image). While this could superficially suggest that representations from the decoder stage onward are less informative than a simple average image, visual inspection of the corresponding reconstructions in Figure 3 reveals the contrary: Despite the higher reconstruction error, these reconstructed images preserve structured, object-specific content, demonstrating that the underlying representations clearly encode meaningful information beyond what is captured by the mean image.

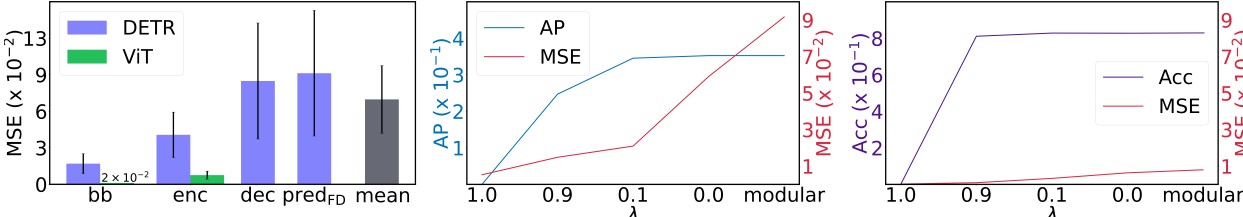

Figure 4: **Left**: Average reconstruction error across processing stages for DETR and ViT on the COCO validation set. *mean* denotes the reconstruction error between images and the dataset mean image. **Center**: Reconstruction loss (MSE) at the DETR decoder stage versus object detection performance (AP), evaluated across different values of $\lambda$. *modular* indicates our modular feature inversion approach without fine-tuning. **Right**: Reconstruction loss (MSE) at the ViT encoder stage versus classification accuracy (Acc) across varying $\lambda$.

Our preliminary analysis suggests a desirable property of our modular feature inversion approach, possibly absent in its classic variant: Despite the potential for error accumulation across sequential inverse modules, our approach produces reconstructions that indicate stage-specific transformations. This observation motivates a more systematic evaluation of the validity of the modular inversion framework, presented next.

## 4.2 Validating Modular Approach

We validated our modular feature inversion approach by comparing it with classic end-to-end feature inversion on image reconstruction quality and computational efficiency. To both ends, in addition to the inverse modules, we trained classic inverse networks for each processing stage of interest in DETR, ViT, Swin, and DeiT III. For a fair comparison, we built these inverse networks by concatenating the inverse modules described in Section 3.3, but trained them end-to-end (from the respective intermediate stage to the input stage) rather than in a modular fashion, see Appendix B.2 for details. As an additional control experiment, we assessed the extent to which reconstructed images reflect the information processing of the TVMs.

### 4.2.1 Image Quality

In Table 1, we report multiple metrics for image reconstructions obtained using modular and classic feature inversion across DETR, ViT, Swin, and DeiT III. We consider the pixel-level metrics MSE, structural similarity index measure (SSIM (Zhou Wang et al., 2004)), peak signal-to-noise ratio (PSNR), the perceptual metrics Learned Perceptual Image Patch Similarity (LPIPS (Zhang et al., 2018)) and Fréchet inception distance (FID (Heusel et al., 2018)), CLIPScore (Radford et al., 2021), and task-specific performance metrics, namely, COCO-style average precision (AP) for DETR and top-1 accuracy for ViT, Swin, and DeiT III.

Across all architectures and processing stages, the classic approach consistently outperforms the modular approach on pixel-level metrics. For perceptual metrics, this trend persists for ViT and DeiT III. In contrast, for Swin, the modular approach achieves better performance at later stages. For DETR, the modular approach yields consistently better FID scores across all stages. This shift becomes more pronounced for semantic metrics. For DETR, the modular approach outperforms the classic approach across all stages, both in terms of CLIPScore and, more substantially, AP. For Swin, a similar pattern emerges in later stages, where the modular approach achieves a higher CLIPScore and significantly better accuracy. In contrast, for ViT and DeiT III, the classic approach remains slightly stronger. However, the gap is considerably smaller than for pixel-level and perceptual metrics, with the modular approach achieving comparable and, in the case of the ViT and DeiT enc stage, equal accuracy. Overall, across all architectures, we observe that the modular

Table 1: Quantitative comparison of reconstruction quality for modular (m) and classic (c) inversion across model stages. Results are reported for DETR on the COCO validation set, and for ViT, Swin, and DeiT III on the ImageNet-1k validation set. Values denote mean $\pm$ standard deviation, except for FID and task performance metrics. CLIPScore is computed using CLIP ResNet-101. Task performance is evaluated on reconstructed images: Average Precision (AP; IoU = 0.50:0.95, maxDets = 100) for DETR, and top-1 accuracy (Acc) for ViT, Swin, and DeiT III. Best values per stage and metric direction are highlighted in bold. For the first processing stage (bb), modular and classic inversion are identical; therefore, only a single value is reported.

| Stage | MSE ($\times 10^{-2}$) $\downarrow$ | SSIM $\uparrow$ | PSNR $\uparrow$ | LPIPS $\downarrow$ | FID $\downarrow$ | CLIPScore $\uparrow$ | AP/Acc $\uparrow$ |
|---|---|---|---|---|---|---|---|
| **DETR** | | | | | | | |
| bb | $1.86 \pm 0.90$ | $0.50 \pm 0.16$ | $17.8 \pm 2.2$ | $0.58 \pm 0.08$ | $110.6$ | $0.77 \pm 0.05$ | $0.05$ |
| enc (m) | $4.60 \pm 2.39$ | $0.45 \pm 0.15$ | $13.8 \pm 2.0$ | $0.63 \pm 0.08$ | $\mathbf{114.2}$ | $\mathbf{0.75 \pm 0.05}$ | $\mathbf{0.05}$ |
| enc (c) | $\mathbf{4.28 \pm 2.03}$ | $\mathbf{0.45 \pm 0.15}$ | $\mathbf{14.1 \pm 2.0}$ | $\mathbf{0.63 \pm 0.08}$ | $134.5$ | $0.72 \pm 0.05$ | $0.02$ |
| dec (m) | $9.37 \pm 5.36$ | $0.37 \pm 0.14$ | $10.9 \pm 2.3$ | $0.70 \pm 0.08$ | $\mathbf{156.1}$ | $\mathbf{0.71 \pm 0.05}$ | $\mathbf{0.03}$ |
| dec (c) | $\mathbf{6.41 \pm 2.75}$ | $\mathbf{0.41 \pm 0.15}$ | $\mathbf{12.3 \pm 1.9}$ | $\mathbf{0.65 \pm 0.07}$ | $309.5$ | $0.67 \pm 0.04$ | $0.00$ |
| pred (m) | $10.12 \pm 5.73$ | $0.36 \pm 0.13$ | $10.6 \pm 2.3$ | $0.72 \pm 0.08$ | $\mathbf{165.3}$ | $\mathbf{0.70 \pm 0.05}$ | $\mathbf{0.01}$ |
| pred (c) | $\mathbf{6.93 \pm 2.86}$ | $\mathbf{0.41 \pm 0.15}$ | $\mathbf{12.0 \pm 1.9}$ | $\mathbf{0.66 \pm 0.07}$ | $345.6$ | $0.66 \pm 0.05$ | $0.00$ |
| **ViT** | | | | | | | |
| bb | $0.02 \pm 0.01$ | $0.97 \pm 0.02$ | $38.3 \pm 3.0$ | $0.03 \pm 0.02$ | $0.4$ | $0.97 \pm 0.02$ | $0.84$ |
| enc (m) | $0.94 \pm 0.55$ | $0.45 \pm 0.11$ | $20.2 \pm 2.2$ | $0.54 \pm 0.06$ | $43.6$ | $0.80 \pm 0.06$ | $\mathbf{0.60}$ |
| enc (c) | $\mathbf{0.75 \pm 0.51}$ | $\mathbf{0.64 \pm 0.15}$ | $\mathbf{22.2 \pm 2.7}$ | $\mathbf{0.38 \pm 0.09}$ | $\mathbf{34.8}$ | $\mathbf{0.85 \pm 0.05}$ | $0.60$ |
| **Swin** | | | | | | | |
| bb | $0.02 \pm 0.01$ | $0.97 \pm 0.04$ | $38.3 \pm 2.2$ | $0.03 \pm 0.02$ | $0.4$ | $0.97 \pm 0.02$ | $0.83$ |
| $s_0$ (m) | $0.03 \pm 0.02$ | $0.96 \pm 0.05$ | $36.9 \pm 2.1$ | $0.04 \pm 0.02$ | $0.5$ | $0.97 \pm 0.02$ | $0.83$ |
| $s_1$ (c) | $\mathbf{0.01 \pm 0.01}$ | $\mathbf{0.99 \pm 0.09}$ | $\mathbf{44.3 \pm 3.0}$ | $\mathbf{0.01 \pm 0.01}$ | $\mathbf{0.1}$ | $\mathbf{0.99 \pm 0.01}$ | $\mathbf{0.84}$ |
| $s_2$ (m) | $0.10 \pm 0.06$ | $0.89 \pm 0.06$ | $30.6 \pm 2.4$ | $0.14 \pm 0.04$ | $4.1$ | $0.96 \pm 0.03$ | $\mathbf{0.81}$ |
| $s_2$ (c) | $\mathbf{0.03 \pm 0.03}$ | $\mathbf{0.96 \pm 0.02}$ | $\mathbf{35.6 \pm 3.2}$ | $\mathbf{0.04 \pm 0.02}$ | $\mathbf{1.1}$ | $\mathbf{0.97 \pm 0.02}$ | $0.80$ |
| $s_3$ (m) | $0.78 \pm 0.57$ | $0.59 \pm 0.15$ | $22.0 \pm 2.8$ | $\mathbf{0.40 \pm 0.07}$ | $\mathbf{36.1}$ | $\mathbf{0.86 \pm 0.06}$ | $\mathbf{0.66}$ |
| $s_3$ (c) | $\mathbf{0.78 \pm 0.50}$ | $\mathbf{0.60 \pm 0.16}$ | $\mathbf{22.0 \pm 2.8}$ | $0.44 \pm 0.10$ | $54.8$ | $0.80 \pm 0.05$ | $0.49$ |
| $s_4$ (m) | $2.91 \pm 1.70$ | $0.39 \pm 0.16$ | $16.0 \pm 2.4$ | $\mathbf{0.56 \pm 0.07}$ | $\mathbf{85.5}$ | $\mathbf{0.78 \pm 0.06}$ | $\mathbf{0.46}$ |
| $s_4$ (c) | $\mathbf{1.78 \pm 1.05}$ | $\mathbf{0.47 \pm 0.17}$ | $\mathbf{18.2 \pm 2.5}$ | $0.61 \pm 0.11$ | $106.5$ | $0.73 \pm 0.07$ | $0.17$ |
| **DeiT III** | | | | | | | |
| bb | $0.01 \pm 0.01$ | $0.98 \pm 0.01$ | $40.4 \pm 2.8$ | $0.02 \pm 0.02$ | $0.2$ | $0.98 \pm 0.01$ | $0.82$ |
| enc (m) | $1.12 \pm 0.42$ | $0.36 \pm 0.10$ | $19.7 \pm 1.6$ | $0.54 \pm 0.07$ | $41.9$ | $0.83 \pm 0.06$ | $0.67$ |
| enc (c) | $\mathbf{0.48 \pm 0.30}$ | $\mathbf{0.70 \pm 0.13}$ | $\mathbf{24.0 \pm 2.7}$ | $\mathbf{0.32 \pm 0.08}$ | $\mathbf{21.8}$ | $\mathbf{0.87 \pm 0.05}$ | $\mathbf{0.67}$ |

approach yields stronger perceptual and semantic metrics relative to the classic approach when pixel-level fidelity is already low, i.e., in deeper stages of Swin and across all stages of DETR. In contrast, for ViT and DeiT III, no such advantage is observed, as both architectures maintain comparatively high pixel-level fidelity across all stages.

### 4.2.2 Computational Efficiency

We compared the computational efficiency of our modular feature inversion approach to the classic feature inversion approach in terms of parameter count and training time (see Table 2). In addition to ViT and DETR, we include results for Swin. Because ViT and DeiT III are architecturally equivalent, their parameter counts and training speeds are equal.

Table 2: Computational efficiency of classic and modular feature inversion. We report parameter count (#Params, in millions) and training time (min/epoch).

| Model | #Params (M) | | Time (min/epoch) | |
|---|---|---|---|---|
| | Classic | Modular | Classic | Modular |
| DETR | 358 | **98** | 114 | **51** |
| ViT / DeiT III | 87 | **86** | 473 | **377** |
| Swin | 149 | **87** | 550 | **245** |

For the modular method, the total number of parameters corresponds to the sum of parameters of the inverse modules. In contrast, in the classic approach, the total number of parameters is the sum of parameters across all inverse networks that invert the forward path from the different processing stages of interest.

To compare training time, we trained the inverse components of both approaches within a shared pipeline, following standard deep learning practices. All experiments were conducted on a single NVIDIA A100 GPU. We used the COCO 2017 dataset for DETR and ImageNet-1K for ViT and Swin, applying the same normalization and resizing augmentations as in the original training procedures of the respective models. Batch sizes were set to the largest power of two that satisfied memory constraints (32 for DETR, 64 for ViT, and 128 for Swin). We employed the Adam optimizer with default hyperparameters, tuning only the learning rate. For both approaches, we first computed intermediate representations at the processing stages of interest using the forward model. In the modular approach, we then reconstructed intermediate representations and computed losses as defined in Equation (1). In contrast, in the classic approach, the input image was reconstructed directly from all intermediate representations, with losses computed directly at the image level. All parameters were then updated jointly.

As expected from our theoretical analysis in Section 3.3, our data show that modular feature inversion requires fewer parameters than classic feature inversion across all architectures. For DETR, the modular approach reduces the parameter count by approximately 73%, and for Swin by about 42%, while for ViT the parameter counts are roughly equal. The larger efficiency gains for DETR and Swin compared to ViT can be attributed to two factors. Firstly, we consider a larger number of processing stages of interest for DETR and Swin. Secondly, the initial processing stage in ViT is relatively lightweight, as its bb is implemented as a simple linear projection. These differences in parameter count are also reflected in training time.

We emphasize that the reported parameter counts and training times for both approaches could be reduced through careful tuning and should therefore be interpreted as empirical observations rather than strict benchmarks. Furthermore, we report the training time in minutes per epoch. However, the total number of epochs needed for training varies across inverse modules and inverse networks. Notably, many inverse modules and inverse networks continued to improve marginally even after a significant training length, making it difficult to define a clear convergence point. In practice, we trained most inverse modules for approximately 100 epochs, though we observed that the inverse modules converged slightly faster than the classic end-to-end inversion networks.

### 4.2.3 Indicativeness of Reconstructions

DNNs for vision tend to progressively abstract and discard image details during processing. Consequently, image reconstruction, particularly from deeper stages, requires inferring or compensating for missing information. Due to the nature of the MSE objective used during training, this process favors the average of plausible solutions rather than a specific instance. To evaluate whether this averaging yields meaningful reconstructions for TVMs, we progressively inject reconstruction-derived information into the forward pass of DETR and ViT, and assess whether the resulting reconstructions become correspondingly sharper, i.e., exhibit reduced averaging. To this end, we trained end-to-end inverse networks along finetuning DETR and ViT variants for image reconstruction. Specifically, we combined the reconstruction loss with the objectives of the respective architecture $L_{\mathrm{OBJ}}$, following the approach of Rathjens & Wiskott (2024):

$$L(\theta \cup \phi_{j:0}) = \lambda L_{\mathrm{MSE}}(\theta) + (1 - \lambda)L_{\mathrm{OBJ}}(\theta) + L_{\mathrm{MSE}}(\phi_{j:0}) \tag{3}$$

Here, $\theta$ denotes the parameter of the forward network (DETR or ViT), $\phi_{j:0}$ denotes the parameters of the inverse network. We trained four model variants with $\lambda \in \{0.0, 0.1, 0.9, 1.0\}$ and set $j$ to dec for DETR and enc for ViT, see Appendix B.3 for details. Notably, $\lambda = 0.0$ corresponds to the classic feature inversion approach, where the forward weights $\theta$ are not influenced by the reconstruction loss.

Figure 5 presents the image reconstructions obtained with fine-tuned inverse models alongside those generated using our modular approach and the classic feature inversion approach. Across all examples, a consistent pattern emerges: For high $\lambda$ values, the reconstructions maintain high fidelity, capturing image details accurately. As $\lambda$ decreases, reconstruction quality deteriorates. This effect is particularly pronounced in DETR, especially in the last two columns, where blur is high. In contrast, ViT exhibits this effect to a lesser degree. Interestingly, for DETR, clear differences emerge between the last two columns: The reconstructions in the $\lambda = 0$ column, which represent the classic feature inversion approach, exhibit a grayish tone, whereas those in the modular approach column display more saturated colors.

We quantitatively analyzed image reconstructions in Figure 4, which displays the MSE alongside AP for DETR and MSE alongside accuracy for ViT. The results align with the qualitative assessment of the reconstructed images: As $\lambda$ decreases, reconstruction error increases. Moreover, as $\lambda$ decreases, the object detection and classification performances of DETR and ViT improve, highlighting a trade-off between reconstruction quality and the tasks of the architectures. Unsurprisingly, the MSE is lower for the classic feature inversion approach than for our modular approach.

Taken together, our results reveal a reasonable averaging in the reconstruction process. Importantly, the grayish tone observed in reconstructions produced by the classical approach helps explain why it achieves lower pixel-level error but poorer semantic fidelity: The classic approach tends to fill in missing image information with dataset-level averages, resulting in gray, structureless regions that minimize MSE but lack semantic meaning. In contrast, the modular approach, due to the independent training of its components, must compensate for missing information already at intermediate stages. As a result, it is encouraged to reconstruct plausible content within more abstract representations, leading to semantically more meaningful outputs.

In conclusion, comparing classical and modular feature inversion reveals that our approach offers significant advantages for TVMs. It is computationally more efficient and can produce semantically more coherent reconstructions when many processing stages are involved or when the model exhibits a high degree of abstraction and omission of information details.

### 4.3 Analyzing Color

Having established the feasibility and validity of our modular inversion framework, we now turn to its primary purpose, i.e., interpreting intermediate representations in TVMs.

Motivated by our preliminary observations indicating substantial differences in color processing between DETR and ViT, we conducted a systematic analysis to investigate these differences in greater detail. To this end, we selectively recolored objects in input images and evaluated how these perturbations affect

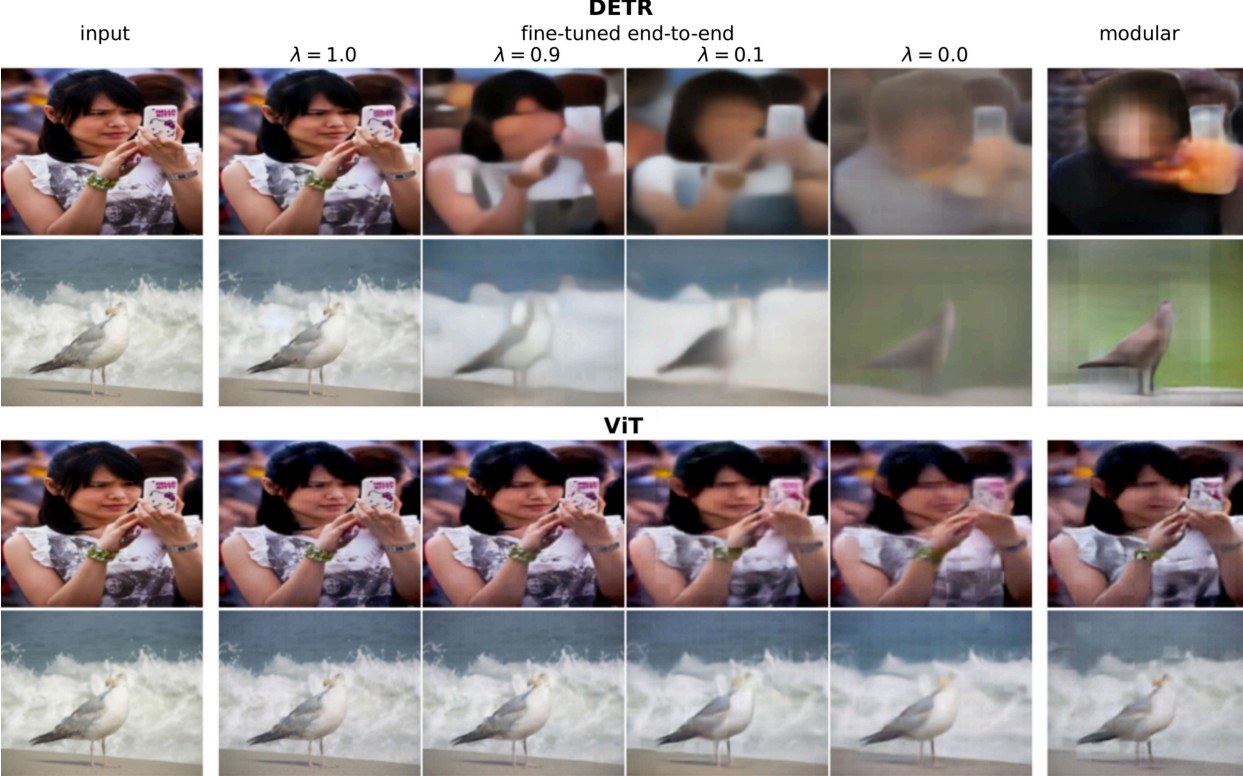

Figure 5: Reconstructions with fine-tuned models. Columns 2–4 show reconstructions from models fine-tuned with different $\lambda$ values. Column 1 displays input images, and column 6 shows reconstructions using our modular feature inversion approach. **Top**: Images from the DETR dec processing stage. **Bottom**: Images from the ViT enc processing stage.

reconstructions across different processing stages. We use COCO segmentation annotations to isolate object instances and apply six color transformations in HSV space: setting the hue to red, green, or blue, rotating the hue by 120° or 240°, and converting the image to grayscale. Figure 6 shows recolored inputs and their corresponding reconstructions.

For DETR, we observe that color perturbations are preserved in image reconstructions from the backbone representations for all objects and filters but gradually fade or disappear almost entirely in image reconstructions from the encoder representations. In $\hat{\mathbf{x}}_{dec:0}$ practically no color perturbation remains. Instead, colors shift toward prototypical representations (red for the stop sign and bus, brown for the bear, red or yellow for the apples, and yellow for the giraffe) even when color information was deleted (see giraffe). In contrast, we do not observe a similar effect in ViT, as color perturbations remain visible in image reconstructions from all processing stages.

We quantified the response to color perturbations by computing the average pairwise MSE between image reconstructions of differently perturbed images from each processing stage. Specifically, given an image $\mathbf{x}_0$, we applied each color filter separately, generating six perturbed versions. We calculated the average pairwise MSE between these perturbed images at the input stage. Similarly, for the backbone stage, we computed the average pairwise MSE between the six corresponding reconstructed images $\hat{\mathbf{x}}_{bb:0}$, following the same approach for the encoder and decoder stages. The left plot in Figure 7 presents these MSE values, averaged across all categories and images in the dataset.

For DETR, we observe that the average pairwise MSE decreases progressively from $\mathbf{x}_0$ to $\hat{\mathbf{x}}_{enc:0}$, indicating increasing similarity. However, at the decoder stage, the MSE returns to input levels. This observation aligns with our qualitative analysis, confirming that reconstructions tend to converge to the same or similar colors

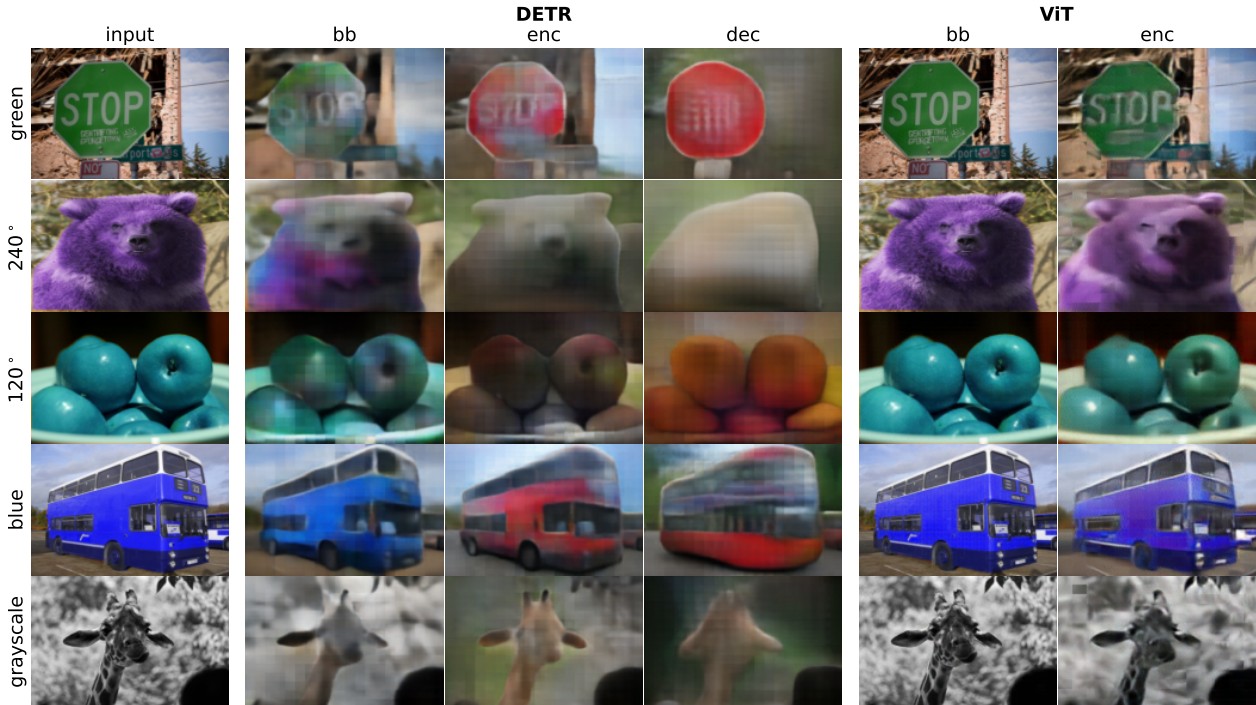

Figure 6: Effects of color perturbations. Rows show images where specific object categories were color-perturbed (from top to bottom: stop sign colored green, bear with colors rotated by 240°, apple with colors rotated by 120°, bus colored blue, giraffe converted to grayscale). Columns 2-4 and 5-6 show reconstructions from different processing stages of DETR and ViT, respectively.

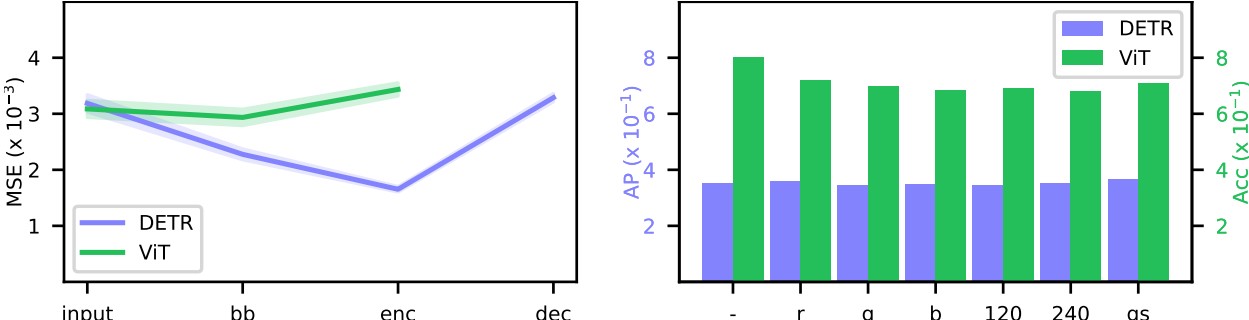

Figure 7: Quantification of color perturbations. **Left**: Average pairwise MSE between image reconstructions of differently perturbed versions of an image, comparing inputs and reconstructions across processing stages. Shaded area indicates 95% confidence intervals over the COCO test set. **Right**: DETR's and ViT's sensitivity to color perturbations (none, red, green, blue, 120° shift, 240° shift, grayscale) in relation to the performance of their objectives.

as they progress through the DETR architecture. The increase in average pairwise MSE at the decoder stage is likely not due to color divergence but rather distortions in object shapes. For ViT, the preservation of color perturbations throughout the architecture is reflected in an almost constant pairwise MSE across processing stages.

The greater loss of color information in DETR compared to ViT suggests that DETR is more robust to color changes than ViT. We tested this hypothesis by evaluating the performance of each architecture on recolored images (see right plot of Figure 7). Specifically, we recolored entire images from the ImageNet dataset and measured classification performance for ViT, as segmentation data was not available. To ensure

a fair comparison, we also applied full-image recoloring for DETR. The results show that accuracy of ViT drops compared to the default setting, whereas DETR remains unaffected.

## 4.4 Analyzing Structure

Another indication from our preliminary analysis is that DETR progressively alters image structure across its processing stages, whereas ViT largely preserves it. To investigate this phenomenon in greater depth, we again reconstructed images from various stages of both architectures, this time focusing on the analysis of structural changes. Given the particularly interesting behavior observed in DETR, we employed two variants for its $\text{pred}^{-1}$: a standard $\text{pred}^{-1}_{\text{FD}}$, which receives $\mathbf{x}_{\text{pred}}$ with full distribution of class logits as input, and an additional $\text{pred}^{-1}_{\text{OH}}$, which takes a one-hot encoded variant of $\mathbf{x}_{\text{pred}}$. The latter retains only the highest-confidence class per detected object, discarding information about the uncertainty in class predictions. Bounding boxes are retained in both variants. We hypothesized that the full distribution of class confidences, beyond the top prediction, encodes meaningful visual cues. The one-hot variant thus enables us to assess more prototypical reconstructions of objects, while preventing the model from exploiting uncertainty and low-confidence class associations during reconstruction.

Figure 8 displays exemplary image reconstructions. For DETR, low-level structural information is generally well-preserved in reconstructions from $\mathbf{x}_{\text{bb}}$. Notably, at the later stages starting from dec, objects undergo significant alterations, including changes in size, shape, structure and orientation (e.g., the person in the first row appears taller with a lowered hand, the sunflowers in the third row shift into a generic green plant, and the horse in the fourth row is reoriented to face right), the addition of contextual elements (left person in the second row appears to be wearing a suit in reconstructions from dec and pred, inferred from the presence of a tie), or complete omissions of objects (e.g., the bollards in the second row, or the photo frame on the wall in the third row are completely abstracted out). Furthermore, some artifacts are introduced, e.g., in the reconstructions from $\text{pred}_{\text{OH}}$ in the fourth row, a dark object appears near the horse that seems to be another person, which is not present in earlier stages. These transformations appeared repeatedly across diverse samples and object classes, suggesting that the model learns structured abstraction behaviors that are consistent within each class.

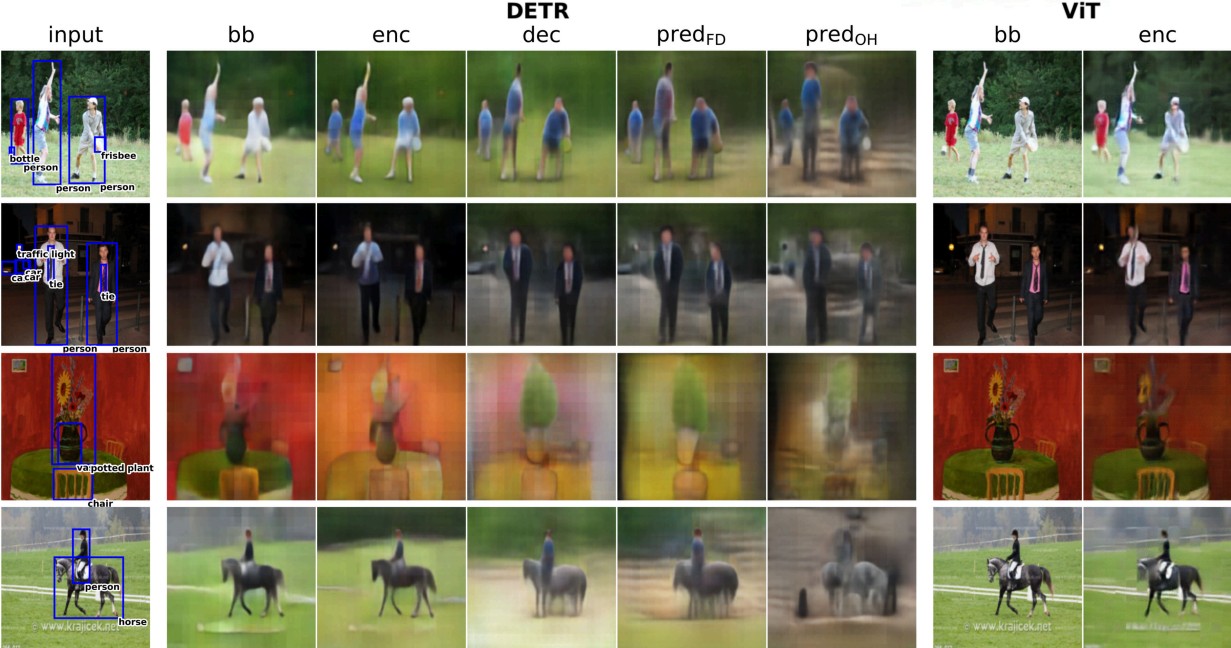

Figure 8: Structural transformation analysis in DETR and ViT

In contrast, ViT reconstructions show little structural change across stages. Object shape, spatial configuration, and contextual elements are consistently preserved, suggesting that ViT retains low-level visual and

semantic information without applying the same degree of abstraction or progression towards prototypical representations as DETR.

We interpret these observations as follows. At higher processing stages, DETR tends to omit image details that are not relevant to object detection, such as objects that are not explicitly recognized (e.g., the omission of bollards and photo frame, compared to detected objects indicated by bounding boxes in the input images). Instead of preserving raw image details, DETR represents objects in a prototypical manner, discarding information deemed irrelevant for recognition, such as pose and shape variations or orientation changes (e.g., the altered posture of the person or the transformed sunflowers). Additionally, DETR appears to learn priors about object co-occurrences and typical scene compositions. It may modify contextual elements to enhance object recognizability, as seen in the addition of a suit coat to emphasize the tie. Using only top-scoring classes for reconstructions can also lead to semantically relevant hallucinations, such as a person appearing near a horse, underscoring the role of model confidence in activating the co-occurrence of related objects. On the other hand, ViT does not appear to undergo these abstractions, as image details remain largely preserved throughout its architecture.

## 4.5 Analyzing Spatial Correlations

Throughout our experiments, we observed that ViT preserves image details more effectively across processing stages than DETR. A plausible explanation for this behavior is that ViT maintains a strong spatial correspondence between image patches and encoder tokens, whereas DETR distributes information from image patches across multiple tokens. To examine this hypothesis, we replaced 20% of randomly selected tokens at two processing stages with identical uniform noise and analyzed the resulting image reconstructions. Specifically, we manipulated $\mathbf{x}_{\text{bb}}$ and generated reconstructions $\mathcal{N}_{\text{bb}:0}^{-1}(\mathbf{x}_{\text{bb}})$ and $\mathcal{N}_{\text{enc}:0}^{-1}(\mathcal{N}_{\text{bb:enc}}(\mathbf{x}_{\text{bb}}))$, which we refer to as $\text{bb}_{\text{man}}$ and $\text{enc}_{\text{man}+}$. Additionally, we manipulated $\mathbf{x}_{\text{enc}}$ to generate reconstruction $\mathcal{N}_{\text{enc}:0}^{-1}(\mathbf{x}_{\text{enc}})$ which we refer to as $\text{enc}_{\text{man}}$.

Our rationale for the experimental setup is as follows: In DETR, at one of its standard image resolutions at $640 \times 480$, each token at the bb stage corresponds to a $20 \times 15$ non-overlapping image patch. For ViT, each token represents a $16 \times 16$ non-overlapping image patch at its standard image resolution $224 \times 224$. If manipulating tokens affects image reconstructions only at their corresponding spatial locations while leaving the remainder of the image unchanged, it would suggest a strong spatial correspondence between tokens and image patches throughout the encoder. Conversely, if token manipulations influence regions beyond their respective spatial locations, it would suggest that the spatial correspondence is relaxed during processing.

To enable a more isolated analysis of spatial correspondences in enc for DETR, we introduced a local inverse backbone variant. In this configuration, each image patch is reconstructed only from a token corresponding to its spatial location, ensuring that the backbone can not globally aggregate local image information.

Figure 9 shows results for two example images for both architectures. For the DETR setup with the standard $\text{bb}^{-1}$, token manipulation leads to slightly increased blurring and color shifts in all image reconstructions. However, the reconstructions do not reveal which tokens were manipulated, as the noisy tokens were consistently filled in with plausible content.

With the local inverse backbone variant for DETR, overall reconstruction quality deteriorates significantly, as expected. Unlike the standard inverse backbone, reconstructed images with the local version enable a more accurate identification of the manipulated tokens. Since the local $\text{bb}^{-1}$ reconstructs each image patch using only a single token, and all manipulated tokens are replaced with identical noise, the corresponding patches appear visually identical, as visible in the $\text{bb}_{\text{man}}$ setup.

In the $\text{enc}_{\text{man}}$ setup, manipulated tokens can still be identified, as their corresponding reconstructed patches differ from those based on unmanipulated tokens. However, these differences are less pronounced, suggesting that manipulated tokens integrate some information from surrounding tokens during processing. In the $\text{enc}_{\text{man}+}$ condition, patches reconstructed from manipulated tokens are nearly indistinguishable from others, as the noisy tokens blend seamlessly into the overall image.

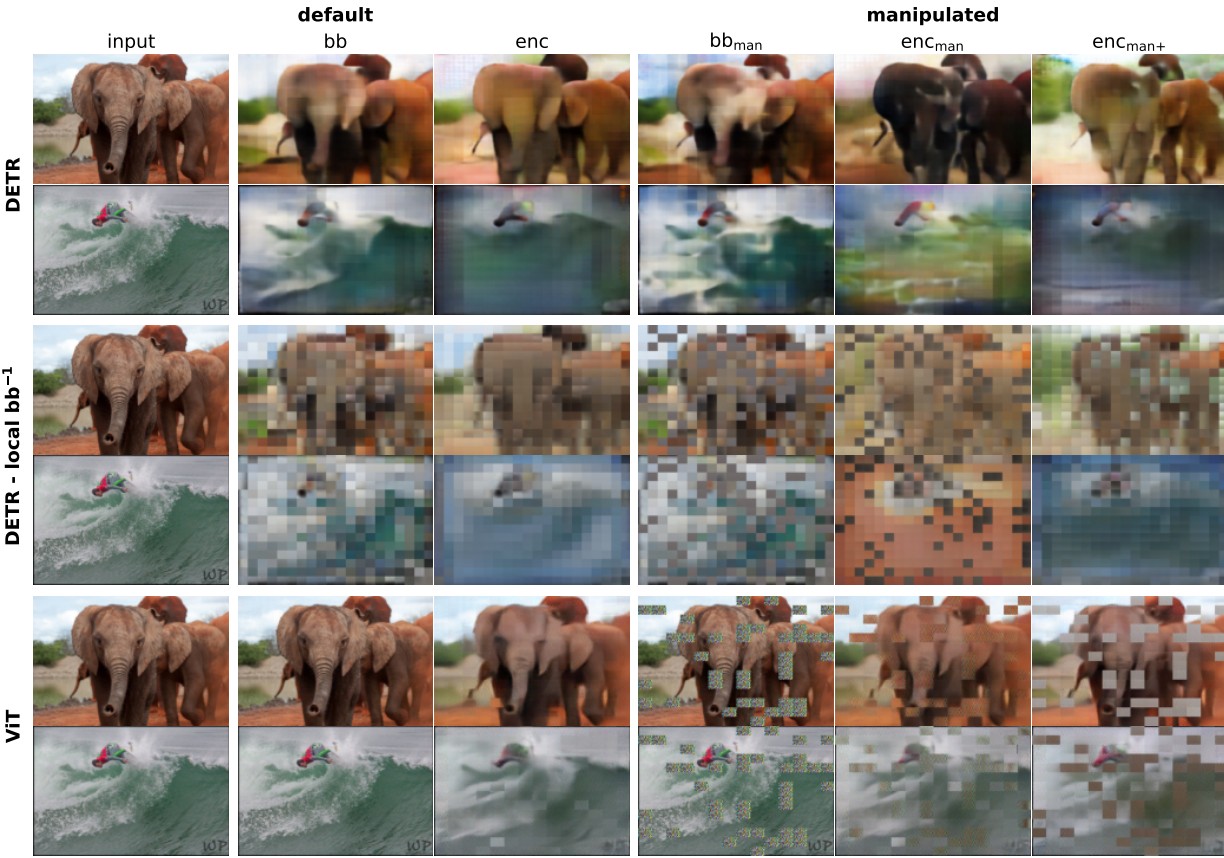

Figure 9: Image reconstructions from default (unmanipulated) and manipulated representations with DETR, a DETR variant with a local inverse backbone, and ViT. Tokens were replaced with the same random uniform noise at different processing stages before reconstruction. For $enc_{man+}$, embeddings were manipulated at the backbone stage, processed through enc, and then used for reconstruction.

The appearance of reconstructed images from manipulated representations in DETR stands in sharp contrast to those obtained from ViT. For ViT, manipulated tokens manifest as visible noise within their corresponding patches, while unmanipulated patches remain unaffected.

The differences in image reconstructions from manipulated representations strongly support the hypothesized distinction in information processing between the two architectures. While both the DETR backbone and encoder distribute image details associated with a given location across multiple tokens, ViT modules preserve a spatial correspondence between tokens and image locations. As a result, the inverse modules in ViT do not require global integration to reconstruct the image, an effect particularly evident in the $enc_{man+}$ setup, where the manipulated tokens remain clearly identifiable despite being processed through multiple stages.

The lower level of abstraction in ViTs suggests that, during the forward pass, greater emphasis is placed on interactions between the class token and image tokens than on self-attention among image tokens. To test this hypothesis, we turned off self-attention in enc, retaining only cross-attention from the class token, and fine-tuned the model on ImageNet-1K. The modified model still achieves approximately 69% top-1 accuracy, supporting our hypothesis. A detailed description of the modified ViT and its training is provided in Appendix B.6.

## 4.6   Analyzing Detection Errors in DETR

Our method also enables a visual inspection of detection errors by examining the reconstructed images to find out how DETR encodes, or fails to encode, objects across stages. As illustrated in Figure 10, objects that are ultimately not detected (e.g., the bicycle in the second row or the potted plants in the third row) are gradually suppressed across the processing stages. Although clearly visible in the input, these elements begin to fade in the reconstructions from bb and enc, and leave no trace in the reconstructions from dec or pred. This gradual disappearance suggests that the model deems them irrelevant and filters them out during object query formation or matching.

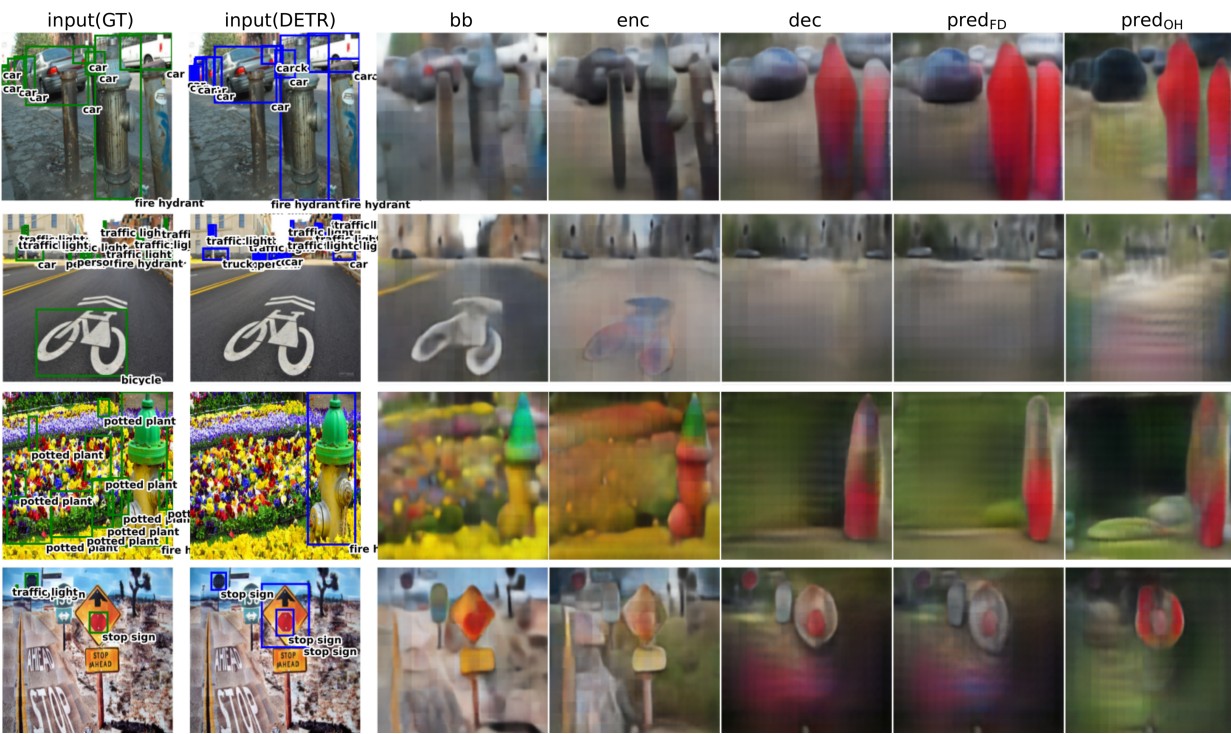

Figure 10: Image reconstructions from various processing stages of DETR. The first and second columns depict input images along with ground truth labels and predictions of DETR, respectively.

In contrast, false positives often exhibit the opposite behavior: Reconstructions from later stages reveal a shift toward features associated with incorrect classes (e.g., the second fire hydrant in the first row or the second stop sign in the fourth row). This suggests that, if DETR misinterprets certain features or contextual cues, it constructs coherent features and consolidates them into prototypical object representations. A contingency-table analysis of 200 randomly sampled detection errors, reported in Appendix A.4, provides additional support for these qualitative trends.

These observations provide a visual trail of where detection errors arise by revealing the stages in the processing pipeline where critical information is lost or misrepresented. This stage-wise visual access to internal representations makes reconstruction-based analysis a valuable diagnostic tool for interpreting the inner workings of DETR, highlighting where in the architecture corrective refinements might be most effective.

## 4.7   Analyzing Intermediate Layers

Until now, we have applied feature inversion only to representations from selected processing stages, thereby excluding layers not explicitly chosen for our analysis. However, unlike CNNs, TVMs, except for Swin, offer a unique analytical opportunity: The intermediate representations within both the encoder and decoder maintain a consistent shape across layers. This property allows intermediate encoder representations to be

passed through the inverse backbone and inverse encoder, and intermediate decoder representations through the inverse decoder, without additional training, even though these modules were not trained for this purpose. Leveraging this unique property, we explored whether our inverse modules could reconstruct images from intermediate encoder and decoder representations, despite the mismatch in training context. Specifically, we analyzed intermediate representations from the encoder and decoder of DETR, as well as from the encoder of ViT. Figure 11 provides illustrative examples.

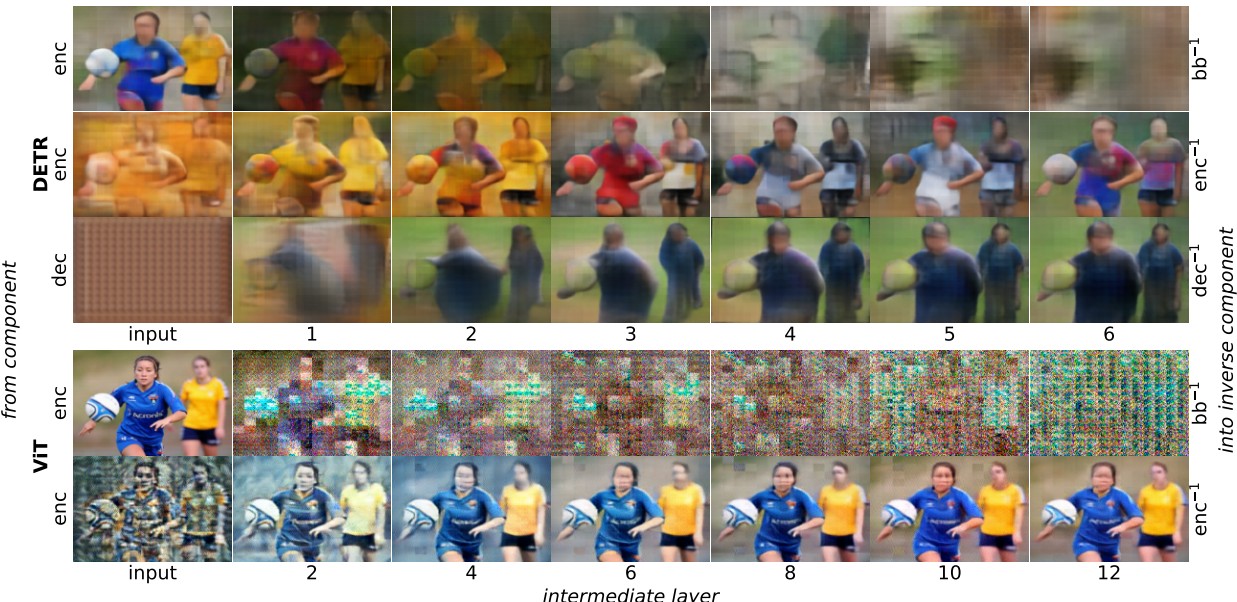

Figure 11: Image reconstructions from intermediate encoder and decoder layers. The left y-axis labels indicate the module from which an intermediate representation was extracted. The x-axis denotes the specific layer in that module from which this intermediate representation was extracted. The right y-axis labels indicate into which inverse module this intermediate representation was fed.

Predictably, for intermediate representations of both architectures, we obtained best reconstruction performances for the representations the inverse modules were trained on: $\mathbf{x}_{bb}$ for $bb^{-1}$, $\mathbf{x}_{enc}$ for $enc^{-1}$ and, for DETR, $\mathbf{x}_{dec}$ for $dec^{-1}$. The quality of reconstructions gradually decreases as we move farther away from the representations the inverse modules were trained on, a pattern particularly evident for the inputs to $dec^{-1}$ since decoder tokens initially hold values that are independent of the input image.

Despite of this degradation, image features are generally preserved across intermediate layers, especially when feeding intermediate encoder representations into $enc^{-1}$. For DETR, most variations in reconstructions from $bb^{-1}$ and $enc^{-1}$ appear as color shifts, whereas reconstructions from $dec^{-1}$ exhibit greater stability in color than in shape. For ViT, reconstructions from $enc^{-1}$ preserve both shape and color, while from $bb^{-1}$ they display strong tiling effects, likely due to the local operations of the inverse backbone. Nevertheless, both color and shape remain discernible. The overall stability of reconstructions across layers is noteworthy, as inverse modules might be expected to produce only noisy outputs when applied to intermediate representations they have not been trained on.

From these observations, we draw three key conclusions. Firstly, the difference in feature preservation between DETR and ViT further highlights their distinct approaches to information abstraction, as DETR progressively alters colors throughout its hierarchy. Secondly, intermediate representations in TVMs evolve gradually across layers, as suggested by Raghu et al. (2021) for ViT. Finally, feature inversion is particularly well-suited for TVMs, as inverse modules can be applied across multiple layers, eliminating the need to train a separate inverse module for each layer.

### 4.8 Disentangling the Origins of Prototypical Representations

To identify the primary cause of the divergent representational behaviors observed between DETR and ViT, we systematically controlled for dataset and augmentation, task, and architectural factors.

One hypothesis is that the observed differences arise from differences in training data. The ViT model analyzed in this work was pretrained on the large-scale JFT-300M dataset, comprising approximately 18,000 classes (Sun et al., 2017), whereas DETR was trained on COCO (Fleet et al., 2014), which contains approximately 90 object categories. The greater visual diversity in JFT-300M may require ViT to preserve finer image details, whereas DETR, trained on a smaller dataset, may afford more abstraction. To test this hypothesis, we controlled for dataset effects by training inverse models on DeiT III, which is architecturally equivalent to ViT but trained on ImageNet-1K with a different augmentation strategy. The resulting reconstructions in Figure 12 remain consistent with ViT, preserving substantially more fine-grained details than DETR. Hence, these findings suggest that the observed divergence may not be explained by dataset-related factors. We therefore examined this question further through the next hypothesis.

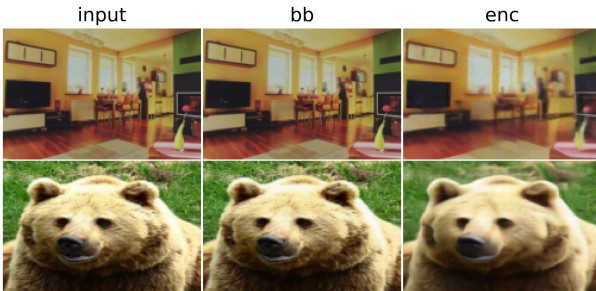

Figure 12: Image reconstructions from processing stages of DeiT III

An alternative hypothesis is that the observed differences stem from the task, i.e., object detection versus image classification. DETR is trained for multi-object detection, requiring both recognition and localization of multiple objects per image, while ViT is trained for image-level classification, where only a single object needs to be recognized without localization. Classification, therefore, may place less demand on the model to transform input features extensively. In contrast, multi-object detection may require more abstract, context-aware representations to support the recognition and spatial localization of all objects.

To test this hypothesis, we adapted DETR for image classification on ImageNet-1K while keeping its architecture largely unchanged, using a single object query for prediction, see Appendix B.4 for details. Figure 13 shows that, despite the change in task and dataset, the reconstructions retain the characteristic DETR behavior, namely a stage-wise progression toward stronger abstraction and increasingly prototypical representations (e.g., altered colors of the canoe and clothing, or changes in the panda's size and orientation; persons are abstracted out at the decoder stage as they are irrelevant for classification). This result indicates that neither the dataset nor the task is the primary driver of the observed differences.

Having ruled out data and task, we examined the role of architectural differences, particularly the backbone, which represents the largest difference between the two architectures. DETR in our experiments uses a CNN (ResNet-50) backbone, which is known to produce progressively more abstract representations (Mahendran & Vedaldi, 2014), whereas ViT employs an invertible linear embedding that preserves all input information. Consequently, the transformer encoders in DETR and ViT operate on substantially different input representations from the outset, potentially leading to distinct information processing throughout the transformers.

To assess the role of the backbone, we progressively truncated the ResNet-50 backbone used in DETR by removing higher residual blocks, resulting in three DETR variants trained for classification, namely a model with the full backbone, a variant with three blocks, and a variant with two blocks, see Appendix B.5 for details. Using our modular feature inversion approach, we analyzed how reconstructions evolve across processing stages for each variant. As shown in Figure 14, this ablation leads to a gradual reduction in

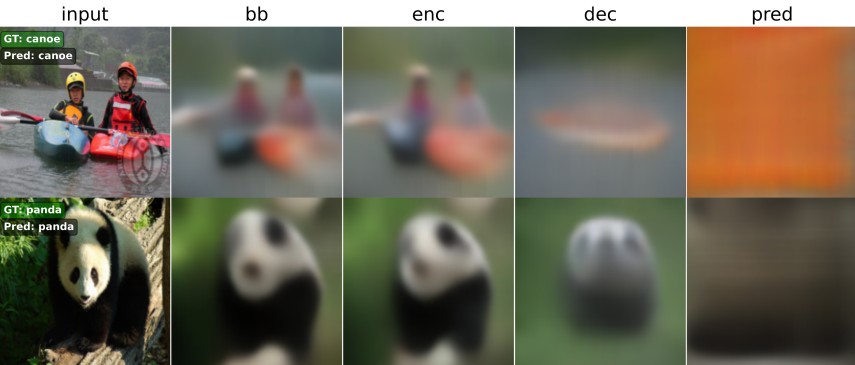

Figure 13: Image reconstructions from the processing stages of DETR trained for image classification on ImageNet-1K. The first column shows the input images together with the ground truth label and the prediction by DETR.

prototypical transformations and greater preservation of fine-grained visual details as blocks are removed from the backbone (e.g., the colors, orientation, and shape of the minibus and the wolf become progressively more faithful to the original image as blocks are removed). In the variant with two blocks, which is closest to the linear backbone in ViT, the reconstruction behavior also became closest to that of ViT, particularly in preserving fine-grained details. The lower reconstruction quality at the decoder stage in this variant likely reflects its substantial accuracy drop of approximately 15% relative to the full ResNet variant. This consistent trend suggests that the backbone is a major contributor to the divergence between the two models.

A plausible interpretation for the significant role of the backbone in shaping the observed representational dynamics is that the DETR encoder receives representations that have already been substantially abstracted by the ResNet backbone. Attention then operates on features with a stronger semantic similarity structure and amplifies these similarities further, resulting in a drift toward prototypical representations. In contrast, when the encoder is provided with less abstract and more instance-specific representations, fewer such similarities are available for amplification, and the resulting inversions preserve more fine-grained details.

Taken together, these experiments suggest that the representational differences between DETR and ViT are not primarily driven by training data or by the distinction between detection and classification objectives. Instead, they point to architectural differences, particularly the backbone and the level of abstraction it provides to the transformer, as major factors. The final representational behavior, however, should be understood as the result of both backbone representations and subsequent transformer processing.

## 5 Discussion

In this work, we set out to apply an efficient variant of the classic feature inversion approach from Dosovitskiy & Brox (2016) to study the intermediate representations of TVMs, focusing on DETR and ViT. We began by formulating a modular version of feature inversion that significantly improves efficiency by replacing large global inverse networks with lightweight, local inverse modules, thereby substantially reducing the number of trainable parameters.

After a preliminary analysis of image reconstructions from DETR and ViT obtained using our approach, we first validated its feasibility for network interpretability of TVMs. To this end, we qualitatively and quantitatively compared our approach to classic feature inversion on DETR, ViT, Swin, and DeiT III, and additionally evaluated DETR and ViT variants fine-tuned for image reconstruction. Our results show that reconstructions obtained via modular feature inversion reflect the underlying processing mechanisms of TVMs. In particular, the method proves most beneficial for architectures with many processing stages and/or those that progressively discard image details. In these settings, modular inversion is significantly more computationally efficient than the classic approach and yields more semantically coherent image reconstructions. However, for architectures with only a few processing stages of interest, the computational

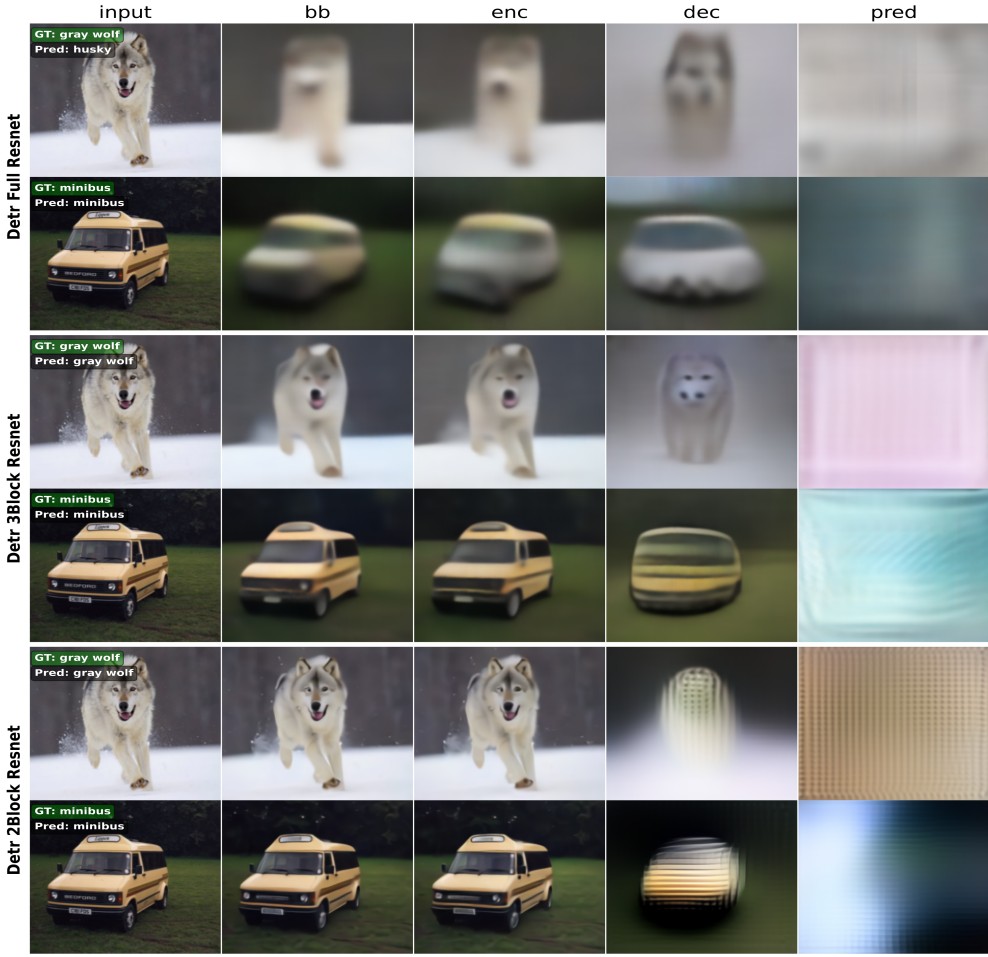

Figure 14: Comparison of stage-wise reconstructions for three DETR variants trained for classification, with full, three blocks, and two blocks of ResNet-50 backbone.

benefit of our method is marginal. In addition, for architectures that preserve image details throughout processing, our method does not produce semantically more coherent images. Nevertheless, even in settings where the modular approach provides little or no benefit, we observed no disadvantage in comparison to classical feature inversion.

Building on this foundation, we showcased the types of systematic interpretability analyses that our method supports, starting with an investigation of how color information is processed in DETR and ViT. We observed that DETR progressively shifts object colors toward prototypical representations, while ViT preserves original color information throughout. Consistent with these findings, DETR shows strong robustness to color perturbations, whereas the classification performance of ViT degrades, challenging previous claims (Naseer et al., 2021; Paul & Chen, 2022) that ViT is remarkably resilient to image perturbations.

We continued our interpretability study with a focused analysis of how image structure is processed in the two architectures. We found that DETR abstracts object structure and context, modifying shapes and poses, and omitting irrelevant features while adding contextually relevant ones, reflecting a shift toward prototypical representations that likely simplify object detection in later stages. In contrast, ViT retains object geometry and spatial layout with minimal distortion, pointing to a lower level of abstraction and a stronger preservation of visual detail.

We then turned towards analyzing spatial correlations between intermediate representations and input images. Using a novel analysis method in the context of feature inversion, specifically, injecting noise into

intermediate representations, we found that ViT encodes spatial information in a localized manner. At the same time, DETR diffuses spatial information more globally. The spatial correspondence in ViT questions the importance of self-attention within the architecture, particularly given that we achieved reasonable classification accuracy in a ViT with disabled self-attention. Notably, Jaegle et al. (2021) have shown that a transformer-based model can achieve competitive accuracy on ImageNet-1k using only cross-attention. However, in their model, self-attention was still applied to register tokens, and its computational complexity exceeded that of ViT.

After briefly showcasing how DETR reconstructions vary with detection errors, we leveraged a key property in many transformer architectures, namely, the constant shape of intermediate representations across encoder and decoder layers. This consistency allowed us to feed these representations into inverse components optimized for reconstruction from different layers. We found that both DETR and ViT refine their representations gradually across layers. This pattern is consistent with prior findings for ViTs (Raghu et al., 2021) and transformer-based large language models (Liu et al., 2023), and we extend it to DETR here.Together, these findings suggest that gradual representational refinement may be a shared characteristic of TVMs and, more broadly, transformers in general. This property also enhances the efficiency of feature inversion in such models.

We concluded the results section by analyzing potential drivers for the divergent representational behaviors between DETR and ViT. After ruling out training data, data augmentation, and task objectives, we identified the backbone architecture as one of the primary factors. In DETR, the CNN backbone already produces comparatively abstract representations before the transformer encoder, which appears to induce a shift in subsequent processing toward more prototypical representations. In contrast, the transformer encoder of ViT operates on less abstract, patch-level image representations and does not exhibit the same drift toward such prototypical abstractions. This conclusion should not be interpreted as isolating the backbone as the only cause. Further controlled variants, such as DETR with a ViT backbone, ViT-style models using the same CNN backbone as DETR, or ViT variants equipped with a detection decoder, would help to further disentangle the contributions of backbone design and subsequent transformer processing stages.

From a methodological perspective, we have shown that modular feature inversion is both more efficient and more naturally aligned with the architecture of TVMs than the classic approach. These properties make it well-suited for analyzing modern iterations of TVMs such as DINOv2 (Oquab et al., 2024) or SAM (Kirillov et al., 2023). Furthermore, since our approach is not limited to TVMs and we expect it to offer advantages for a broad range of DNNs, it may also prove valuable for analyzing modern CNN-based models such as ConvNeXt V2 (Woo et al., 2023) or YOLOv8 (Ultralytics).

One particularly intriguing property of our method in the case of DETR and Swin is that, despite yielding higher image reconstruction error than classic feature inversion, images are better suited for network interpretability. While we can attribute this effect to the closer mirroring of the forward processing path obtained with modular feature inversion compared to the classic approach, its extent remains unclear. Future research could further explore this by systematically varying the number of inverse modules and examining their impact on reconstruction quality and interpretability.

In this line of research, future work could also address a fundamental limitation of feature inversion: Even with the modular approach, it remains challenging to conclusively attribute specific properties in reconstructed images to individual processing stages. Drawing reliable conclusions typically requires additional quantitative analysis. However, increasing the number of inverse components may offer finer-grained insights and help localize specific representational effects to particular layers.

Our method may have broader applications beyond network interpretability. In the case of DETR, we observed that undetected objects often vanish in reconstructed images, while misclassified objects tend to appear significantly altered. These findings point to a promising direction for applying modular feature inversion to error detection: By comparing reconstructed images to their inputs, discrepancies may serve as indicators of detection failures.

Drawing a parallel to computational neuroscience, prior work has shown that generative models of episodic memory require the integration of both discriminative and generative processes (Fayyaz et al., 2022). Future

models could build on this idea by unifying a TVM and its inverse within a single architecture. Likewise, TVMs may be well-suited for biologically plausible learning systems, as they naturally support local reconstruction losses (Kappel et al., 2023).

For applicants of TVMs, our results suggest that TVMs should not be treated as a homogeneous model class. In particular, the backbone strongly shapes the type of information available to subsequent processing stages. In applications requiring fine-grained visual fidelity, such as medical imaging, architectures that preserve low-level and instance-specific information may be preferable, as this information is diagnostically relevant. Conversely, architectures that progressively abstract away such details may be advantageous in settings where robustness to superficial variation is more important than fidelity to individual image properties, as illustrated by the robustness of DETR to color perturbations.

In summary, we proposed a modular feature inversion framework for TVMs that enables scalable, component-wise interpretability with minimal training overhead. Applied to DETR and ViT, it revealed shared and distinct representational dynamics in abstraction, spatial encoding, and robustness. Beyond interpretability, the approach shows promise for error detection and biologically inspired learning, positioning modular inversion as a practical tool for probing modern vision models and guiding future discriminative-generative integration.

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

# A  Further Results

## A.1  DeiT III

Analogous to the experiments in the main paper, we present reconstructions for standard images (Figure 15), color-perturbed images (Figure 16), and manipulated images (Figure 17) for DeiT III.

Across all reconstruction settings, DeiT III exhibits the same behavior as ViT. This consistency suggests that (a) our method is applicable to TVMs regardless of their specific training schemes, and (b) the detailed reconstructions observed when inverting ViT, compared to DETR, are not merely a consequence of the large-scale dataset used for ViT training.

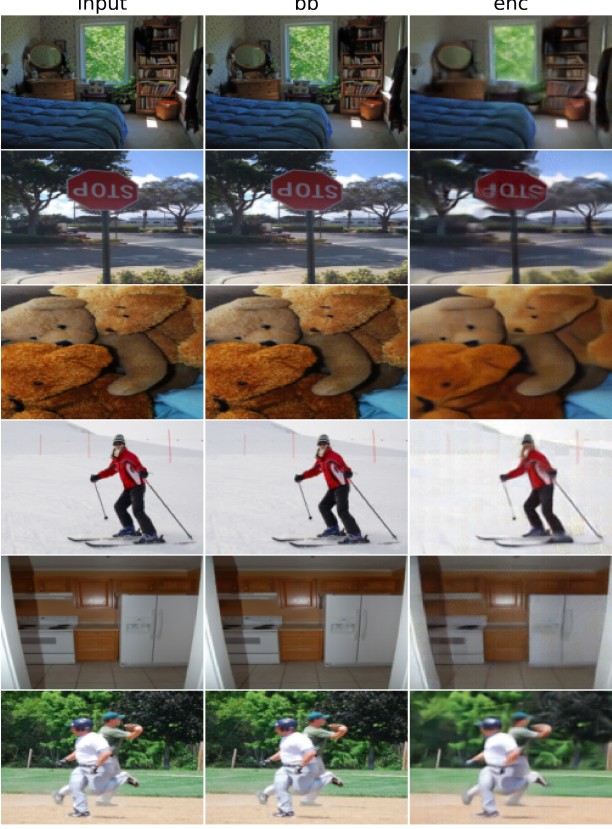

Figure 15: Image reconstructions from various processing stages of DeiT III.

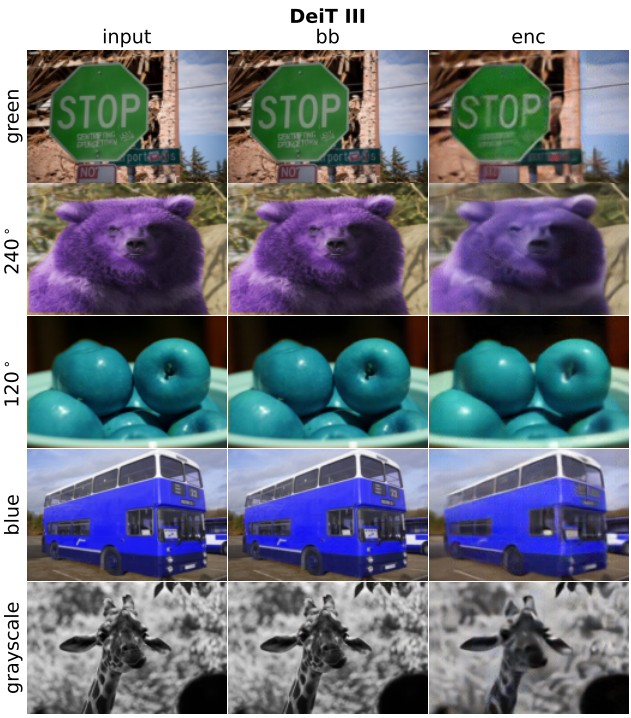

Figure 16: Image reconstructions from various processing stages of DeiT III on color-perturbed images, analogous to Figure 6.

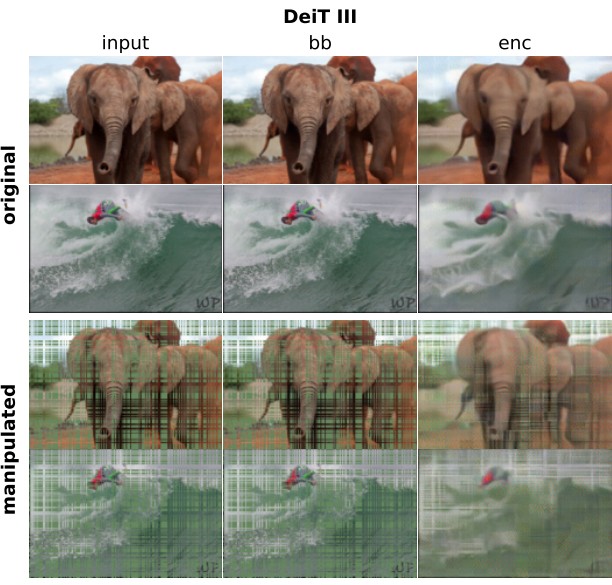

Figure 17: Image reconstructions from various processing stages of Swin on randomly manipulated images, similar to Figure 9.

## A.2 Swin

We present reconstructions for standard images (Figure 18), color-perturbed images (Figure 19), and manipulated images (Figure 20) for Swin.

Across all experiments, the image reconstructions are highly detailed across most processing stages of Swin, and the reconstructions under manipulations and color perturbations are qualitatively similar to those of ViT. Only the reconstructions from the last and second-to-last processing stages exhibit a loss of fine detail, slight color shifts, and distorted object outlines, with these effects being most pronounced in the final stage.

We interpret these results as follows. From our experiments in Section 4.5, we inferred that image details in ViT are propagated locally, suggesting an emphasis on locality in self-attention. This locality, which ViT must learn from large-scale data, is already inductively encoded in Swin through its local windowing mechanism. Consequently, it is unsurprising that Swin behaves similarly to ViT and can be trained on smaller datasets, as the model's architecture inherently enforces the locality that ViT must acquire through data. The loss of image detail in the final layer is likely due to the average pooling operation applied to the output of stage five prior to the classification head.

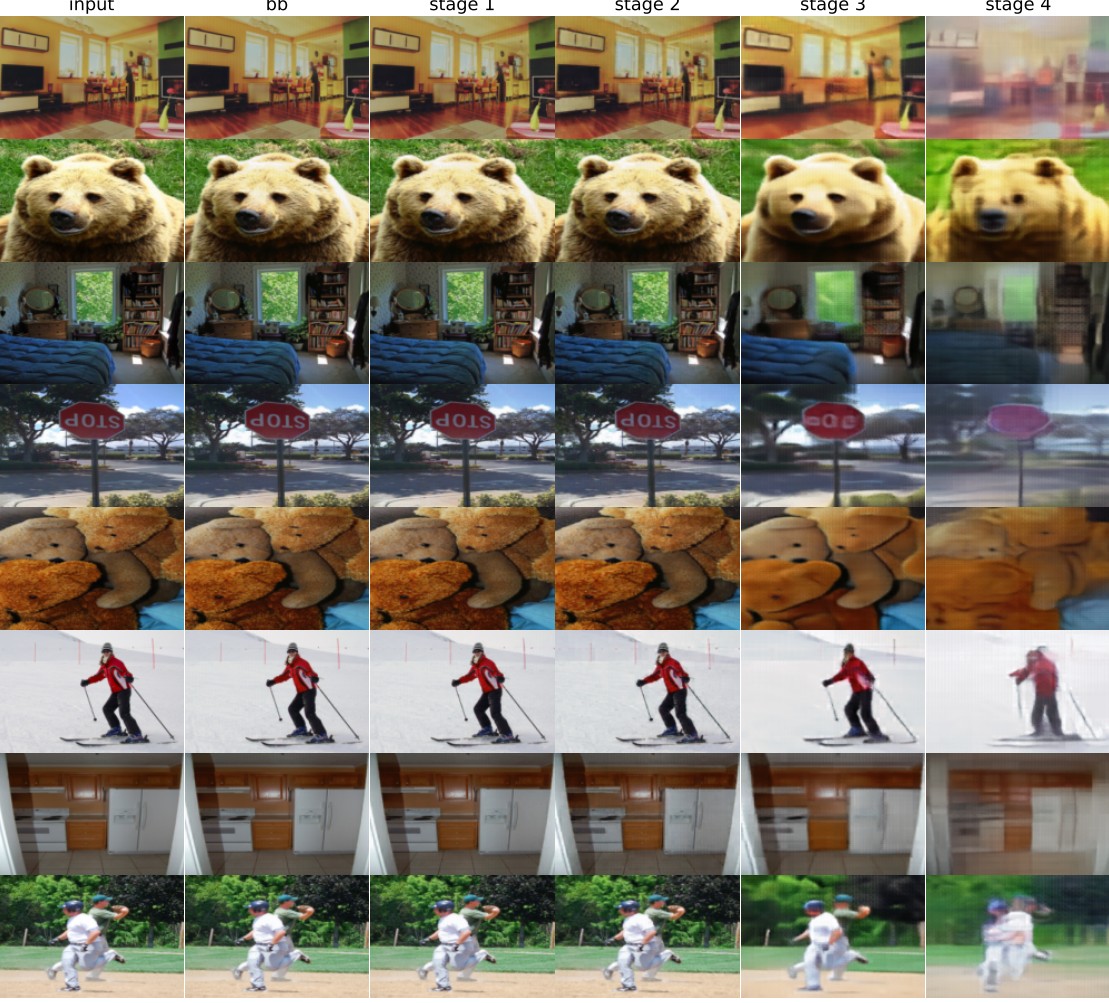

Figure 18: Image reconstructions from various processing stages of Swin.

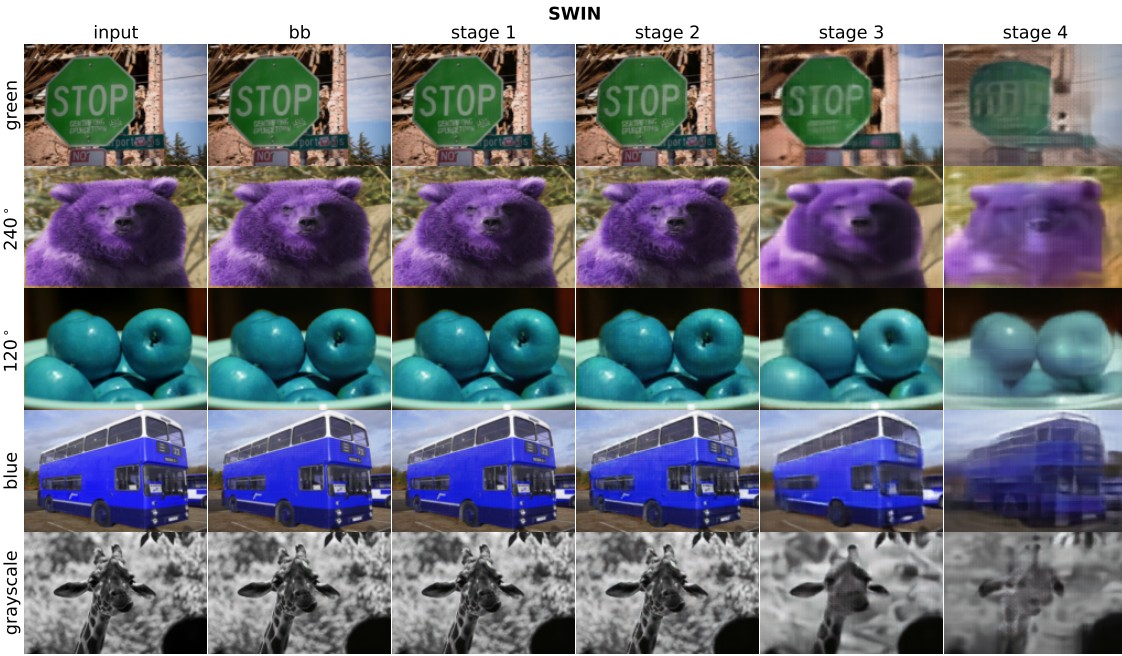

Figure 19: Image reconstructions from various processing stages of Swin on color-perturbed images, analogous to Figure 6.

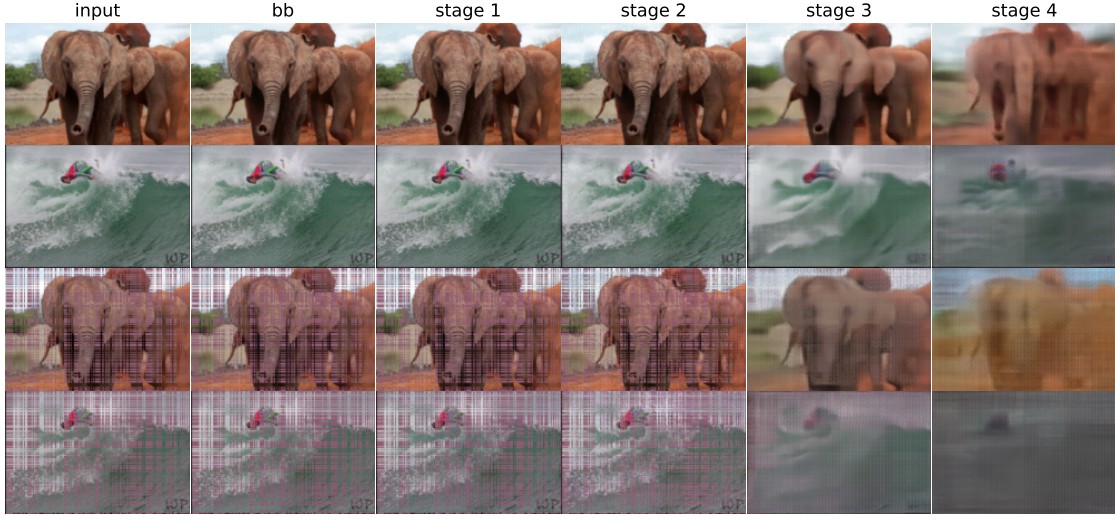

Figure 20: Image reconstructions from various processing stages of Swin on randomly manipulated images, similar to Figure 9.

## A.3   Inversion Bias Control

To control for potential inductive bias introduced by the inversion objective, we repeated the main qualitative analyses from Section 4.3 and Section 4.4 with alternative inverse backbone objectives. Besides the original MSE-trained $bb^{-1}$, we trained variants using SSIM and LPIPS loss functions. We also trained two additional variants combining MSE with SSIM or LPIPS, with equal weighting between the two loss terms.

Figure 21 shows exemplary image reconstructions comparing the three standalone loss objectives. Across all objective configurations, we observed the same qualitative trends. The prototypical color shifts reported in Section 4.3 remains visible (e.g., bus shifting toward red). Similarly, the prototypical shape reconstruction and scene abstraction described in Section 4.4 persisted across the alternative inverse modules (e.g., surrounding scene is abstracted, and the person wearing a tie and vest appears instead as wearing a suit). The combined objective variants (MSE+SSIM and MSE+LPIPS) produced reconstructions that were visually intermediate between those produced by the corresponding individual objectives and showed the same qualitative trends as the standalone objectives.

We further quantified the difference between stage-wise reconstructions produced by different inverse backbone objectives against those of the MSE-trained inverse backbone. As shown in Figure 22, the average MSE between variant objective reconstructions and the MSE-trained baseline remains low across all DETR stages and is substantially smaller than the reconstruction error of the MSE-trained inverse backbone relative to the original images.

While the visual sharpness and texture details varied across objectives, the level of abstraction and the direction of the observed transformations remained consistent. This supports the interpretation that the reported trends primarily reflect DETR's internal representations rather than inductive bias introduced by a specific training objective of the inverse modules.

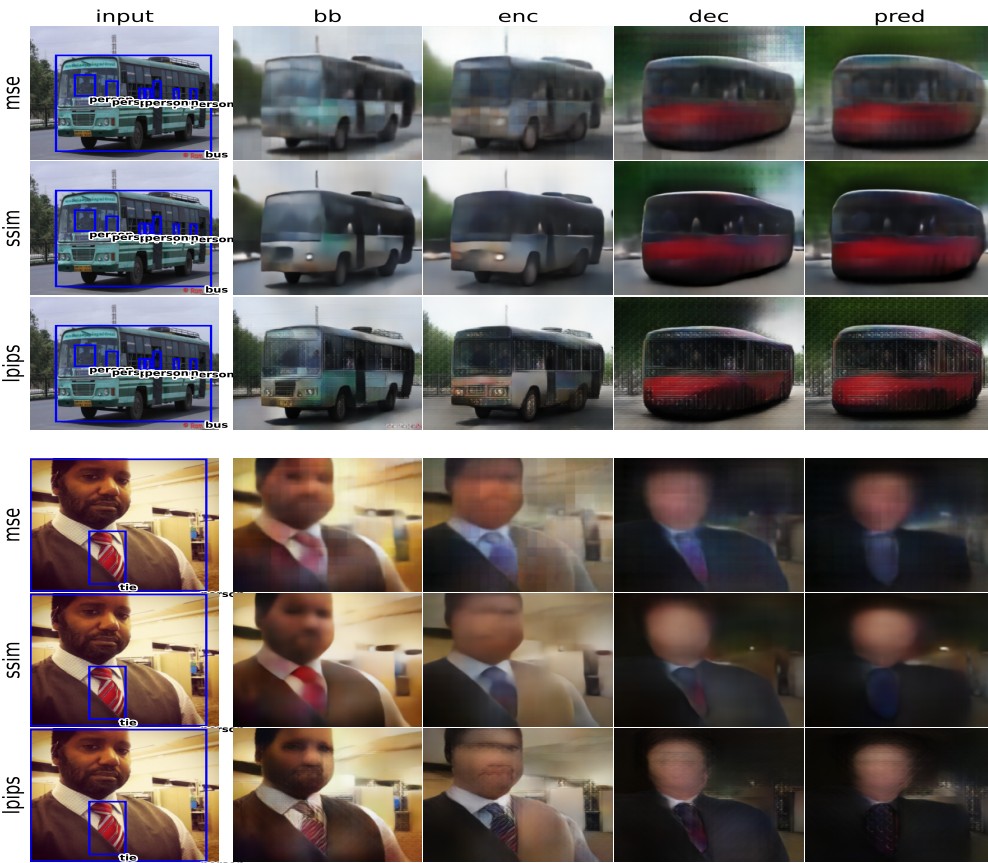

Figure 21: Image reconstructions obtained with inverse backbones trained with different objectives. The figure shows COCO validation examples, reconstructed from successive DETR stages. Rows compare MSE, SSIM, and LPIPS trained inverse backbones, highlighting how the training objective affects reconstructions while preserving the main qualitative trends across the DETR pipeline.

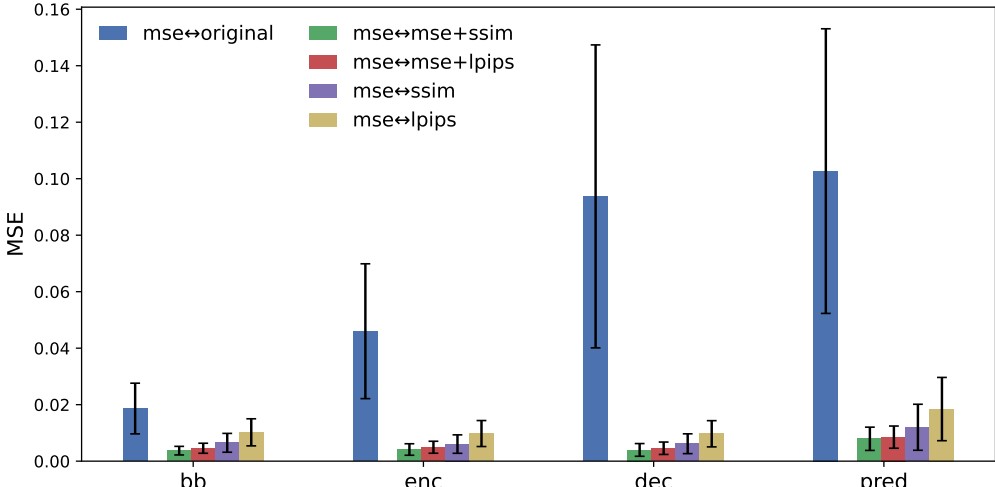

Figure 22: Quantitative analysis of stage-wise reconstruction consistency across inverse backbone objectives on COCO validation set. The blue bar in each stage reports the average reconstruction error of the MSE-trained inverse backbone with respect to the original images. The remaining bars report, for each DETR stage, the difference between reconstructions produced by an alternative objective and those produced by the MSE-trained inverse backbone.

### A.4 Quantification of Detection Errors Traceability

To complement the qualitative analysis in Section 4.6, we quantified the observed error patterns using 200 randomly sampled error cases from the COCO validation set, including 100 false negatives and 100 false positives produced by DETR. Each case was manually assigned to one of three categories. A case was labeled positive when the stage-wise reconstruction made the relevant error pattern visually traceable, namely fading object evidence for false negatives or amplified prototypical object features for false positives. A case was labeled negative when the expected pattern was not observable. Cases were labeled ambiguous when limited reconstruction quality or severe scene clutter prevented a reliable visual judgment. Table 3 reports the resulting counts, which provide quantitative support for the qualitative observations in Section 4.6., indicating that the observed reconstruction patterns are not limited to isolated examples.

Table 3: Quantitative analysis of detection errors traceability

| Error type | Positive | Negative | Ambiguous |
|---|---|---|---|
| False negatives | 92 | 2 | 6 |
| False positives | 97 | 1 | 2 |

## B  Implementation Details

We implemented all models in PyTorch (Paszke et al., 2019), version 2.6.0. For ViT, Swin, and DeiT III, we used the implementations and pretrained checkpoints provided by the timm library (Wightman, 2019), version 1.0.14. For DETR, we used the official Meta Research implementation[2] and pretrained checkpoint as the forward model. Training of all models was performed on NVIDIA A100 GPUs with 40 GB of memory. Full implementation details are provided in the accompanying code repository.[3]

---

[2]https://github.com/facebookresearch/detr
[3]The code is submitted as supplementary material to preserve anonymity during double-blind peer review.

Table 4: Architecture summary of the inverse modules for DETR. Output shapes are reported for a batch size of $b$. Repeated transformer blocks are grouped for compactness.

| Inverse module | Layer / Block | Output shape | Parameters |
|---|---|---|---|
| $\text{pred}^{-1}$ | Linear + ReLU | $[b, 100, 512]$ | 49,664 |
| | Linear + ReLU | $[b, 100, 256]$ | 131,328 |
| | Linear + ReLU | $[b, 100, 256]$ | 65,792 |
| $\text{dec}^{-1}$ | Query embedding | $[100, b, 256]$ | 25,600 |
| | $6\times$ DETR decoder block | $[300, b, 256]$ | 9,472,512 |
| | Self-attention per layer | $[300, b, 256]$ | 263,168 |
| | Cross-attention per layer | $[300, b, 256]$ | 263,168 |
| | Feed-forward network per layer | $[300, b, 256]$ | 1,050,880 |
| $\text{enc}^{-1}$ | $6\times$ DETR encoder block | $[300, b, 256]$ | 7,890,432 |
| | Self-attention per layer | $[300, b, 256]$ | 263,168 |
| | Feed-forward network per layer | $[300, b, 256]$ | 1,050,880 |
| | LayerNorms per layer | $[300, b, 256]$ | 1,024 |
| $\text{bb}^{-1}$ | ConvTranspose2d + BN + ReLU | $[b, 2048, 15, 20]$ | 530,432 |
| | ConvTranspose2d + BN + ReLU | $[b, 1792, 15, 20]$ | 33,035,520 |
| | ConvTranspose2d + BN + ReLU | $[b, 1536, 15, 20]$ | 24,777,216 |
| | ConvTranspose2d + BN + ReLU | $[b, 1024, 30, 40]$ | 14,158,848 |
| | ConvTranspose2d + BN + ReLU | $[b, 512, 60, 80]$ | 4,720,128 |
| | ConvTranspose2d + BN + ReLU | $[b, 256, 120, 160]$ | 1,180,416 |
| | ConvTranspose2d + BN + ReLU | $[b, 128, 240, 320]$ | 295,296 |
| | ConvTranspose2d + BN + ReLU | $[b, 64, 480, 640]$ | 73,920 |
| | ConvTranspose2d + Sigmoid | $[b, 3, 480, 640]$ | 1,731 |
| | **Total trainable parameters** | | **96,408,835** |

We trained all inverse modules for DETR on the COCO 2017 training dataset (Fleet et al., 2014), and all remaining inverse modules and models on the ImageNet-1K training dataset (Krizhevsky et al., 2012). All analyses of reconstructed images were conducted on the corresponding test sets. For the main analyses, we trained one instance of each inverse module.

## B.1 Inverse Modules for DETR, ViT, Swin, and DeiT III

We used the following checkpoints for the TVMs: DETR-R50, ViT-B/16, Swin-B, and DeiT-B. The architectural details of the inverse modules are summarized in Tables 4 to 6. Note that we used the same inverse-module architectures for ViT and DeiT III.

For training, we used a simple training loop with the Adam optimizer and gradient clipping at 1.0 for all inverse modules. To reduce the number of forward passes through the TVMs, all inverse modules associated with a given TVM were trained in parallel using intermediate representations obtained from the same forward pass. Consequently, all inverse modules for a given TVM used the same batch size, which was chosen to maximize GPU memory utilization. We trained all models for 100 epochs, although good performance was achieved within the first 20 epochs for most models, with only minor improvements thereafter. We tuned only the learning rate, selecting the best-performing value from the discrete set $10^{-2}, 10^{-3}, 10^{-4}, 10^{-5}$, as reported in Table 7. We did not use additional training techniques such as learning-rate scheduling or data

Table 5: Architecture summary of the inverse modules for ViT and DeiT III. Output shapes are reported for a batch size of $b$. Repeated transformer blocks are grouped for compactness.

| Inverse module | Layer / Block | Output shape | Parameters |
|---|---|---|---|
| enc$^{-1}$ | 12× ViT transformer block | $[b, 197, 768]$ | 85,026,816 |
| | LayerNorm before attention per block | $[b, 197, 768]$ | 1,536 |
| | Multi-head self-attention per block | $[b, 197, 768]$ | 2,360,064 |
| | LayerNorm before MLP per block | $[b, 197, 768]$ | 1,536 |
| | MLP per block | $[b, 197, 768]$ | 4,722,432 |
| bb$^{-1}$ | ConvTranspose2d + BN + ReLU | $[b, 256, 28, 28]$ | 787,200 |
| | ConvTranspose2d + BN + ReLU | $[b, 64, 56, 56]$ | 65,728 |
| | ConvTranspose2d + BN + ReLU | $[b, 16, 112, 112]$ | 4,144 |
| | ConvTranspose2d + Sigmoid | $[b, 3, 224, 224]$ | 195 |
| | **Total trainable parameters** | | **85,884,083** |

augmentation beyond rescaling and normalization, because this simple setup already yielded satisfactory results and showed no signs of overfitting.

## B.2 Classic Feature Inversion Networks

For training the classic inverse networks, we used the same training setup as for the modules in our modular approach. We kept the batch size fixed and adapted only the learning rate, using the same hyperparameter search procedure. Table 8 reports the selected learning rates for the inverse networks trained to invert from the respective processing stages. As before, all models were trained for 100 epochs, though most achieved good performance within the first 20 epochs, with only minor improvements thereafter.

## B.3 Finetuned Models

For fine-tuning, we initialized the forward models (DETR or ViT) from the same checkpoints used to train the inverse modules, and initialized the inverse networks with the weights of the inverse modules trained as described in Appendix B.1. For each fine-tuned variant, we used AdamW with default parameters and a learning rate of $1 \times 10^{-5}$. In Equation (3), for DETR, we used the Hungarian loss as $L_{\text{OBJ}}$, i.e., the same loss used by Carion et al. (2020), with the same hyperparameters. For ViT, we used cross-entropy loss as $L_{\text{OBJ}}$. We fine-tuned all models for 100 epochs, using COCO 2017 for DETR and ImageNet-1K for ViT.

## B.4 DETR for Classification

To train DETR for classification, we made only minor changes to the architecture. Specifically, we changed the output dimensionality of the class logits in the DETR prediction head to 1000 to account for the difference in the number of classes between COCO 2017 and ImageNet-1K. We initialized the backbone with pretrained ResNet-50 weights and reinitialized all other weights in the architecture. Since the model was trained for classification instead of object detection, we computed the classification loss (cross-entropy) only on the first query-token output of the decoder after passing it through the prediction head. We used only the class logits and did not take the bounding-box output into account.

We trained DETR for image classification on ImageNet-1K for 200 epochs using AdamW with a batch size of 64, a weight decay of $10^{-4}$, and gradient clipping at a maximum norm of 0.1. Learning rates of $10^{-4}$ and $10^{-5}$ were used for non-backbone and backbone parameters, respectively. The learning rate was reduced by a factor of 0.5 after 100 epochs.

We trained the inverse modules for the classification DETR analogously to the inverse modules for standard DETR, as described in Appendix B.1.

Table 6: Architecture summary of the inverse modules for Swin. Output shapes are reported for a batch size of $b$. Repeated Swin Transformer blocks are grouped for compactness.

| Inverse module | Layer / Block | Output shape | Parameters |
|---|---|---|---|
| $s_4{}^{-1}$ | 2× Swin transformer block | $[b, 7, 7, 1024]$ | 25,203,264 |
| | LayerNorm before attention per block | $[b, 7, 7, 1024]$ | 2,048 |
| | Window attention per block | $[b, 49, 1024]$ | 4,203,808 |
| | LayerNorm before MLP per block | $[b, 49, 1024]$ | 2,048 |
| | MLP per block | $[b, 49, 1024]$ | 8,393,728 |
| | PatchSplitting: LayerNorm + Linear | $[b, 14, 14, 512]$ | 2,099,200 |
| $s_3{}^{-2}$ | 18× Swin transformer block | $[b, 14, 14, 512]$ | 56,791,584 |
| | LayerNorm before attention per block | $[b, 14, 14, 512]$ | 1,024 |
| | Window attention per block | $[4, 49, 512]$ | 1,053,328 |
| | LayerNorm before MLP per block | $[b, 196, 512]$ | 1,024 |
| | MLP per block | $[b, 196, 512]$ | 2,099,712 |
| | PatchSplitting: LayerNorm + Linear | $[b, 28, 28, 256]$ | 525,312 |
| $s_2{}^{-2}$ | 2× Swin transformer block | $[b, 28, 28, 256]$ | 1,582,224 |
| | LayerNorm before attention per block | $[b, 28, 28, 256]$ | 512 |
| | Window attention per block | $[16, 49, 256]$ | 264,520 |
| | LayerNorm before MLP per block | $[b, 784, 256]$ | 512 |
| | MLP per block | $[b, 784, 256]$ | 525,568 |
| | PatchSplitting: LayerNorm + Linear | $[b, 56, 56, 128]$ | 131,584 |
| $s_1{}^{-2}$ | 2× Swin transformer block | $[b, 56, 56, 128]$ | 397,896 |
| | LayerNorm before attention per block | $[b, 56, 56, 128]$ | 256 |
| | Window attention per block | $[64, 49, 128]$ | 66,724 |
| | LayerNorm before MLP per block | $[b, 3136, 128]$ | 256 |
| | MLP per block | $[b, 3136, 128]$ | 131,712 |
| $bb^{-1}$ | ConvTranspose2d + BN + ReLU | $[b, 32, 112, 112]$ | 16,480 |
| | ConvTranspose2d + BN + Sigmoid | $[b, 3, 224, 224]$ | 393 |
| | **Total trainable parameters** | | **86,747,937** |

## B.5 Truncated ResNet Backbones

We truncated the ResNet-50 backbone in two ways. For the 3Block variant, we removed the final ResNet block, resulting in an output of size $[b, 1024, 14, 14]$ for input images of size $[b, 3, 224, 224]$. The backbone output was then flattened, permuted, and linearly projected to $[49, b, 256]$ to match the input format of the DETR encoder, where 49 denotes the number of tokens and 256 the token dimension. Analogously, for the 2Block variant, we removed the final two ResNet blocks, yielding an output of size $[b, 512, 28, 28]$ for input images of the same size. Again, we flattened, permuted, and linearly projected the output to $[196, b, 256]$ to match the input format of the DETR encoder.

We trained DETR with truncated ResNet backbones and the corresponding inverse modules, as for the DETR classification model described in Appendix B.4. Specifically, we initialized the weights of the truncated ResNet backbone from ResNet-50, while reinitializing the weights of the transformer encoder, decoder, and classification head.

Table 7: Training hyperparameters for inverse modules.

| Model | Learning rate | Batch size |
|---|---|---|
| $bb^{-1}$ for DETR | 0.001 | 32 |
| $enc^{-1}$ for DETR | 0.001 | 32 |
| $dec^{-1}$ for DETR | 0.001 | 32 |
| $pred^{-1}$ for DETR | 0.001 | 32 |
| $bb^{-1}$ for ViT | 0.001 | 64 |
| $enc^{-1}$ for ViT | 0.0001 | 64 |
| $bb^{-1}$ for Swin | 0.0001 | 128 |
| $s_1^{-1}$ for Swin | 0.0001 | 128 |
| $s_2^{-1}$ for Swin | 0.0001 | 128 |
| $s_3^{-1}$ for Swin | 0.0001 | 128 |
| $s_4^{-1}$ for Swin | 0.0001 | 128 |
| $bb^{-1}$ for DeiT III | 0.001 | 64 |
| $enc^{-1}$ for DeiT III | 0.0001 | 64 |

Table 8: Training hyperparameters for inverse networks.

| Model | Learning rate |
|---|---|
| $bb^{-1}$ for DETR | 0.001 |
| $enc^{-1}$ for DETR | 0.001 |
| $dec^{-1}$ for DETR | 0.000001 |
| $pred^{-1}$ for DETR | 0.000001 |
| $bb^{-1}$ for ViT | 0.001 |
| $enc^{-1}$ for ViT | 0.0001 |
| $bb^{-1}$ for Swin | 0.0001 |
| $s_1^{-1}$ for Swin | 0.0001 |
| $s_2^{-1}$ for Swin | 0.0001 |
| $s_3^{-1}$ for Swin | 0.0001 |
| $s_4^{-1}$ for Swin | 0.0001 |
| $bb^{-1}$ for DeiT III | 0.001 |
| $enc^{-1}$ for DeiT III | 0.0001 |

## B.6  ViT With Disabled Self-Attention

Our ViT variant with disabled self-attention is based on ViT-B/16, the architecture used throughout our analysis. In each transformer block, we replaced standard self-attention with attention only from the class token to the image tokens, analogous to a cross-attention operation. Consequently, image tokens were updated only by the block-wise MLP and normalization operations and did not receive information from other tokens. The class token, in contrast, was updated by the MLP and normalization operations and additionally aggregated information from all image tokens in every block. All other architectural components were left unchanged.

For training, we initialized the model with pretrained weights from the ViT-B/16 checkpoint. To fine-tune the model efficiently on ImageNet-1K, we followed the training recipe of Touvron et al. (2022). The hyperparameters are reported in Table 9 and largely follow those recommended by Touvron et al. (2022) for training ViTs on ImageNet-1K. We did not fully optimize these hyperparameters for our modified architecture, but adapted some of them to match the available computational resources, specifically a single NVIDIA A100 GPU. For hyperparameters not listed in the table, we used the default settings of the timm library (Wightman, 2019). Please refer to our repository for full details.

Table 9: Hyperparameters used for fine-tuning a ViT with disabled self-attention

| Hyperparameter | Value |
|---|---|
| Batch size | 256 |
| Optimizer | LAMB |
| LR | 0.003 |
| Gradient Clip. | 1.0 |
| Epochs | 400 |
| LR decay | cosine |
| Warm-up epochs | 5 |
| Minimum learning rate | $10^{-5}$ |
| Cooldown epochs | 10 |
| Decay epochs | 30 |
| LR Noise percentage | 0.4 |
| LR Noise standard deviation | 0.1 |
| Drop path rate | 0.05 |
| Loss | BCE |
| Repeated augmentation | 3 |
| H. flip | 0.5 |
| RRC | True |
| 3 Augment | True |
| Mixup alpha | 0.8 |
| Cutmix alpha | 1.0 |
| Mixup/CutMix switch probability | 0.5 |
| ColorJitter | 0.3 |

Since the training run was computationally expensive, we trained only a single model. The model reached a best top-1 accuracy of 69.84% after 395 epochs, though 69% accuracy was already reached after 264 epochs. Since we did not fully optimize the hyperparameters, we expect that the model could reach slightly higher accuracy in a more optimized setting, but that the overall gain would likely be limited.

As a control experiment, we also fine-tuned the ViT-B/16 checkpoint with self-attention enabled, using the same training setup but a smaller learning rate of $1 \times 10^{-5}$, as the original learning rate proved unstable. Due to differences in normalization between the ViT-B/16 checkpoint and our setup, accuracy dropped from 84.15% to 80.98% before any fine-tuning was applied. During training, the model reached a peak accuracy of 84.84% after 52 epochs, marginally exceeding the 84.15% accuracy reported for the original ViT-B/16. Accuracy did not improve over the following 30 epochs, while the validation loss increased monotonically, so we stopped training.

