# OpenReview forum: "Understanding Transformer-Based Vision Models via Modular Feature Inversion"
_TMLR — Decision pending for TMLR_

### Review · Reviewer_yseu · 2026-05-17

**Summary Of Contributions:**

The paper proposes a modular variant of feature inversion: rather than training one end-to-end inverse network per probed stage, one inverse module is trained per forward module, and reconstructions are obtained by composing them. Parameter cost becomes roughly constant in the number of probed stages rather than linear. The framework is applied to DETR, ViT, Swin, and DeiT III, with the comparative interpretability study focused on DETR vs. ViT.

The most informative findings are: (i) DETR progressively constructs prototypical objects (shifting colors toward category prototypes, reshaping poses, adding contextually plausible elements, omitting undetected ones), while ViT preserves pixel-level detail throughout depth; (ii) DETR diffuses spatial information across tokens whereas ViT preserves a strong token-to-patch correspondence, demonstrated through a clean token-noise-injection protocol; (iii) a backbone-truncation ablation on DETR (full ResNet-50 → 3 blocks → 2 blocks) shows that the abstraction behavior fades as the backbone shortens, pointing to the CNN backbone rather than data or task as the dominant driver. The cross-layer reuse of inverse modules in §4.7 — feeding intermediate transformer-layer activations into an inverse module trained on a different layer — is a neat, low-cost diagnostic enabled by the modular design.

Strengths. The DETR-vs-ViT comparison surfaces a non-obvious result: the common reading that DETR's transformer encoder is a faithful refinement of backbone features is at least partially wrong — the encoder amplifies abstraction the backbone has begun. The token-noise and backbone-truncation experiments are well-designed for the question they ask.

Weaknesses. The "semantic coherence" advantage of modular inversion is only true for DETR (and late Swin stages); the framing does not reflect this. Several interpretive claims rest on MSE-trained inverse modules without controlling for the inverter's own inductive bias. The §4.5 self-attention ablation is dramatically underspecified for the weight it carries. The §4.8 backbone attribution is supported only in one direction.

**Audience:**

Yes

**Audience Explanation:**

DETR has received considerably less interpretability attention than ViT, and the finding that its transformer encoder amplifies rather than merely refines backbone-induced abstraction — supported by the color, structure, and backbone-truncation experiments — is a substantive contribution to how the community reads detection-transformer internals. The token-noise protocol in §4.5 and the cross-layer inverse-module reuse in §4.7 are also reusable methodological pieces. Even readers who push back on individual interpretive claims will find at least one of these worth citing.

**Broader Impact Concerns:**

None. The work is a diagnostic tool applied to standard pretrained models on public benchmarks, with no novel data collection, deployment claim, or capability frontier shift. A Broader Impact statement is not required.

**Claims And Evidence:**

No

**Claims Explanation:**

1. The §4.5 self-attention ablation is underspecified for the claim it supports. The paper states that disabling self-attention in ViT's encoder and fine-tuning on ImageNet-1K yields "approximately 69% top-1 accuracy," described as "reasonable" and as supporting the hypothesis that class-token-to-image-token cross-attention dominates over self-attention. ViT-B/16 normally reaches 81–84% top-1 on ImageNet-1K depending on pretraining; a 12–15 pp drop is not negligible. The manuscript does not report the fine-tuning recipe, the matched control (the unmodified ViT fine-tuned identically), which layers had self-attention disabled, or whether other components were re-initialized. As written, this single number cannot bear the interpretive load placed on it.

2. Inverse-module inductive bias is a confound that is acknowledged in passing but not controlled. A core interpretive move throughout §§4.3–4.7 is to attribute properties of the reconstruction to properties of the forward representation. But MSE-trained reconstructions are conditional means, and the deconvolutional bb⁻¹ carries its own well-documented artifacts (the tiling effect visible in Figure 11 right, which the paper correctly attributes to the local inverse backbone — and then continues to use as evidence about ViT's spatial encoding in §4.5). The "grayish vs. saturated" contrast in §4.2.3 that justifies preferring modular reconstructions may itself be a property of the inversion procedure (one shot per stage to commit to a sample) rather than evidence about what the forward network encodes. Without re-running the headline qualitative claims under at least one alternative inversion configuration (e.g., MSE + LPIPS loss), the reader cannot separate model behavior from inverter behavior.

3. The §4.8 backbone attribution is one-sided. Truncating DETR's ResNet-50 toward something more ViT-like makes DETR's reconstructions more ViT-like. The converse — attaching a CNN stem to ViT (a ViT-Hybrid checkpoint exists in the original ViT paper) and showing the appearance of prototypical abstractions — is not run. The conclusion that "the backbone is the primary factor" rests on one direction of the symmetric experiment that would actually establish it.

Several other interpretive claims (DETR "adds a suit coat to emphasize the tie," "hallucinates a person near a horse" in §4.4) rest on isolated examples; a small contingency-table count over COCO would convert them from anecdote to evidence, but I treat this as a strengthening point rather than blocking.

**Requested Changes:**

1. Re-frame the "semantic coherence" claim conditionally. The abstract and introduction generalize a property that Table 1 shows holds only for DETR and late Swin stages; for ViT and DeiT III, classic inversion is at least as good on every metric reported. The §4.2.1 narrative already acknowledges this — the front matter and Discussion should too.

2. Either substantially expand the §4.5 self-attention ablation or soften the conclusion drawn from it. At minimum: specify the fine-tuning recipe, report a matched control (unmodified ViT fine-tuned identically), and clarify which encoder blocks had self-attention disabled. If the 69% figure does not survive a fair comparison, the §4.5 hypothesis and the corresponding sentences in the Discussion should be downgraded from "supports our hypothesis" to "preliminary evidence consistent with."

3. Add at least one inverse-module-bias control. Re-run the most load-bearing qualitative claims — DETR's prototypical-color shift (§4.3) and prototypical-shape construction (§4.4) — with a different inverter configuration (e.g., MSE + LPIPS, or a different inverse-head architecture). The goal is not to improve reconstructions but to demonstrate the direction of the interpretive claims is robust to the choice of inverter.

4. Add the symmetric backbone experiment to §4.8. Run inverse modules on the ViT-Hybrid variant (ResNet stem + ViT encoder) from Dosovitskiy et al. (2020), or an equivalent. If the abstraction behavior appears, the "backbone is the primary factor" conclusion is established; if it does not, the conclusion needs to be qualified to "the backbone is a major factor, but the interaction with the transformer encoder matters."

5. Quantify the §4.6 detection-error claims with a small contingency-table count over the COCO validation set: among DETR's false negatives, in what fraction does the object visibly fade in x̂_dec:0? Among false positives, in what fraction do prototypical features visibly appear? Even N=200 would be sufficient to move these claims from anecdote to evidence.

6. Move the per-module architecture and parameter counts, training recipe, and the "at least three instances" protocol (were figures from a fixed seed? are reported metrics averaged over instances?) from the code repository into the paper or a structured appendix.

---

> ### Author Response · Authors · 2026-06-04
> **Requested Changes**
>
> We thank the reviewer for their thorough and constructive review. We uploaded a new version of the manuscript that incorporates the reviewers' requests. For convenience, all changes are written with a blue colored font. Below, we address the raised points and summarize the changes made to our manuscript.
>
> 1. As requested, we refined the wording on semantic coherence in the abstract and discussion of our manuscript.
>
> 2. We added a detailed description of the modified ViT variant and its training to the appendix of the manuscript. In short, in each transformer block, we replaced standard self-attention with class-token-to-image-token attention, analogous to cross-attention, while leaving all other architectural components unchanged. Thus, image tokens were updated only by the block-wise MLP and normalization operations, whereas the class token additionally aggregated information from all image tokens. The model was initialized from a pretrained ViT-B/16 checkpoint and fine-tuned on ImageNet-1K following the training recipe of \citet{touvron_deit_2022}.
> We agree with the reviewer that a $\sim 15\%$ performance gap between the modified ViT variant and the original model is not negligible. We therefore adopted a more cautious formulation. Nevertheless, we still consider the  $\sim 69\%$ top-1 accuracy achieved with disabled self-attention to be noteworthy: While self-attention clearly contributes to performance, the result suggests that a substantial fraction of the predictive capacity of the model is mediated by the class-token-to-image-token attention mechanism.
> As suggested, we also conducted a control experiment in which we fine-tuned ViT-B/16 from the original checkpoint using the same training setup. However, this fine-tuning only marginally improved the accuracy of ViT-B/16 from 84.15\% to 84.84\%. The control experiment is described in more detail in the newly added appendix section.
>
> 3. As suggested, we conducted an additional control experiment to assess whether the most load-bearing qualitative claims about DETR were affected by biases in our inversion process. Specifically, we trained inverse backbones for DETR using similar but slightly different training objectives: LPIPS, SSIM, LPIPS combined with MSE, and SSIM combined with MSE, instead of MSE alone. All control settings revealed the same qualitative pattern as the original objective. We added this control experiment, along with further details and analysis, to the appendix.
>
> 4. We thank the reviewer for suggesting this control experiment. We agree that equipping a ViT with a CNN backbone and observing prototypization would provide strong evidence that prototypization is primarily caused by the backbone. However, the specific control experiment suggested by the reviewer is less straightforward than it may appear at first sight.
> Although the ViT-Hybrid variant in Dosovitskiy et al. (2020) uses a ResNet-50 stem, it takes features from the fourth processing stage of the ResNet, with dimensions [b, 1024, 14, 14] where b denotes the batch size for a standard ViT input image of size [b, 3, 224, 224].
> In contrast, the DETR backbone uses features from the fifth processing stage of the ResNet. For a standard DETR input image of size [b, 3, 480, 640], this stage produces features of size [b, 2048, 15, 20].
> Thus, reconstructions obtained with an inverted Hybrid ViT would not be directly comparable to those obtained with the inverted DETR. Instead, the inverted Hybrid ViT would be more comparable to the DETR three-block ResNet variant discussed in Section 4.8. Notably, the three-block ResNet backbone already displayed substantially less prototypization than the full ResNet backbone, and the prototypization we observed was most pronounced in later DETR processing stages, such as the decoder, which is absent in ViT. We therefore do not expect pronounced prototypization in this ViT-Hybrid setting.
> An alternative experiment would be to construct a Hybrid ViT variant that uses input features from the fifth processing stage of ResNet-50 rather than the fourth. However, this would introduce substantial overhead, as such a model is currently not publicly available and would likely require training from scratch. This, in turn, could be computationally expensive and would require a dedicated investigation of suitable training strategies.
> While such an experiment could be interesting, it is currently beyond the scope of our resources. Instead, following the reviewer’s suggestion, we have weakened our claim regarding the cause of prototypization.
>
>
> 5. As suggested by the reviewer, we added a brief analysis with a contingency table to quantify the traceability of detection errors in DETR to the appendix.
>
>
> 6. As requested, we added detailed descriptions of our models and their training to the appendix and clarified how we generated our results.

---

### Review · Reviewer_Huua · 2026-05-31

**Summary Of Contributions:**

**Summary**

This paper proposes a modular feature inversion framework for analyzing transformer-based vision models. Instead of training a separate end-to-end inverse network for each processing stage, the method trains local inverse modules corresponding to different components of the forward model and chains them to reconstruct images from intermediate representations. The paper applies this framework to several transformer-based vision models, with the main analysis focused on DETR and ViT, and additional results on Swin and DeiT III. The main empirical finding is that DETR progressively abstracts visual information into more prototypical object-level representations, while ViT preserves more fine-grained appearance, color, and spatial detail across depth.

**Strengths**
- The paper presents a clear and useful interpretability framework for probing vision transformer representations.
- The visual analyses cover multiple aspects, including color, structure, spatial correspondence, detection errors, and intermediate layers.
- The additional controlled experiments improve the explanatory depth of the paper, especially by suggesting that the backbone plays an important role in the emergence of DETR-like prototypical representations.

**Weakness**
- The paper remains primarily an interpretability and analysis work. Although the revised version better explains where the observed representational differences come from, it still does not fully show how these insights translate into concrete model-design or training improvements.
- The controlled comparisons are improved but still not really comprehensive. For example, a ViT-style detector, a DETR variant with a ViT backbone, or frozen-vs-finetuned backbone comparisons would further clarify the causal role of architecture, task, and training.
- Some conclusions still rely heavily on qualitative interpretation of reconstructed images, although this issue is somewhat mitigated by the added quantitative metrics and control experiments.

**Audience:**

Yes

**Audience Explanation:**

Yes. I believe this paper would be of interest to researchers working on vision transformers, object detection, representation analysis, interpretability, and model diagnostics.

**Broader Impact Concerns:**

I do not have major broader impact concerns. The paper is primarily an interpretability and analysis work on existing vision models.

**Claims And Evidence:**

Yes

**Claims Explanation:**

I believe the main claims are now mostly supported by clear and convincing evidence. The paper claims that modular feature inversion is more efficient than classical feature inversion, and the revised manuscript now provides quantitative comparisons. This substantially addresses my previous concern that the efficiency advantage was not empirically substantiated.

**Requested Changes:**

- Though the revised experiments improve the explanation of the DETR--ViT difference, but the paper should be careful not to overstate the conclusion. The results suggest that the backbone is an important factor, but other factors such as decoder design, detection training, and pretraining/fine-tuning protocols may still contribute.
- Add a more concrete discussion of what researchers should do differently based on the findings as my main remaining concern is that the paper is still primarily analytical. The discussion would be more useful if it more explicitly connected the findings to practical design decisions.
- Discuss remaining controlled comparisons as limitations because though added controls are valuable, the paper could explicitly mention that further variants, such as DETR with a ViT backbone, would be useful to further separate the roles of backbone, objective, and architecture.

---

> ### Author Response · Authors · 2026-06-04
> **Requested Changes**
>
> We thank the reviewer for their constructive review and for acknowledging the improvements made in the revised version of our manuscript. We have uploaded a new version of the manuscript that incorporates the reviewers’ requests. For convenience, all changes are shown in blue font. Below, we address the points raised and summarize the changes made to the manuscript. Please note that we also made additional changes to the manuscript in response to the comments of Reviewer yseu.
>
> 1. As suggested, we have weakened our claim regarding the backbone as the primary cause of prototypization.
>
> 2. As suggested, we have expanded the Discussion section to elaborate on the practical implications for practitioners working with TVMs. In particular, we argue that TVMs should not be treated as a homogeneous model class and that the application should inform the choice of the specific TVM architecture. However, since the focus of our work is indeed primarily analytical, these implications remain speculative.
>
> 3. As suggested, we expanded the Discussion section to elaborate on additional control experiments. Please also refer to our response to Reviewer yseu, Point 4, regarding a control experiment using a ViT with a ResNet backbone.

---

> > ### Comment · Reviewer_Huua · 2026-06-18
> >
> > Thanks for the revision and response. The revised version looks good to me.
> >
> > Regarding this paper, I noticed that the authors have made substantial revisions based on the feedback from the first submission. In particular, I think they have put considerable effort into addressing the efficiency-related claims and the claim that the observed differences are backbone-driven rather than target-driven, especially in Section 4.8. These revisions are closely related to the suggestions I raised in the first-round review, and I appreciate the authors’ efforts to better support their claims.
> >
> > After reading the other reviewers’ opinions, I also agree with their concerns regarding the generalizability of the paper’s conclusions, such as the discussion on “DETR and ViT” raised by reviewer YFXv and the possible “inverse-module-bias control” issue raised by reviewer yseu. I have also reviewed the authors’ responses to these concerns, and I will continue to monitor how the other reviewers assess these responses.

---

### Review · Reviewer_YFXv · 2026-06-14

**Summary Of Contributions:**

The paper examines feature inversion via inverse networks and introduces a modular version that leads to a more computationally efficient version. The approach is applied to several large-scale transformer-based vision models. The paper reports experimental results that provide insights into how the different architectures handle contextual shape, image detail, and color perturbations.

Strengths

S1.	The paper introduces a novel modular variant for feature inversion that is more parameter and computationally efficient than existing methods.

S2.	The method is applied to four different vision models and the experimentation is thorough, leading to some intriguing insights.

Weaknesses

W1.	Some of the claims of the paper are supported primarily by analysis of a single model, with results provided for a single dataset. It’s not clear how meaningful these results are.

W2.	The paper labels very different components in different architectures with the same nomenclature making the analysis confusing.

W3.	The paper spends considerable effort comparing two architectures that are designed (primarily) for different purposes (detection versus classification). It is not surprising that the behaviour is different, so it’s not clear whether the provided insights are valuable or useful.

**Audience:**

Yes

**Audience Explanation:**

Understanding how vision models behave internally is of interest to many readers and the paper outlines a computationally efficient method that shows promise.

**Claims And Evidence:**

No

**Claims Explanation:**

Claims

C1. The paper introduces an efficient feature inversion method based on modular, independently trained inverse modules.

C2. The paper demonstrates how to use reconstructed images to interpret internal processing mechanisms.

C3. The paper identifies shared properties across multiple architectures.

C4. The experiments reveal systematic differences between the architectures (image detail preservation, abstraction, robustness to colour perturbation).

C5. The approach “can produce semantically more coherent reconstructions, particularly when many processing stages are involved or when the model exhibits a high degree of abstraction and omission of information details”

Claim C1 is supported by the methodological development and the results demonstrating computationally efficient operation. Claim C2 is supported in that the experiments provide some examples of analysis conducted using the reconstructions. The paper doesn’t present a clear recipe for using reconstructions to understand internal mechanisms, however. Claim C3 is weakly supported. The word “shared” does not appear in the experiments section of the paper, so it is unclear exactly which results demonstrate the shared properties. The vague “gradual representation changes across layers” is not a meaningful shared property.  Claim C4 is supported, but the reporting of results in the main body of the paper after the basic validation and computation in Sections 4.2.1-4.2.2 focuses almost entirely on DETR and ViT. Claim C5 is very weakly supported. The claim relies on two figures. In Figure 5, it is challenging to see much difference between the modular and end-to-end approach for ViT, so the claim is based on the results of DETR where the task (detection) is a significant mismatch from the reconstruction objective. It’s not clear how Figure 4 supports the claim since the \lambda=0 performance for AP is very similar to the modular performance. So the claim, which is phrased in a relatively general sense, is supported by one figure that analyzes one architecture on one dataset.

**Requested Changes:**

(1)	Claims C3-C5 should be adjusted or the paper should provide additional experimental support. For C3, if the paper stresses it as a claimed contribution, then the experiments should clearly identify what is shared. This should go beyond “gradual changes” to include something more meaningful. For C4, there should be less emphasis on DETR and ViT. The paper claims to analyze four transformer-based models, but almost all of the diagnostic results address only two. For C5, there should be clearer support for the claim. One figure, for one dataset and one architecture, is not enough to support the generality of this claim.

(2)	DETR and ViT were designed with very different purposes in mind. It’s not clear that revealing differences between how these architectures behave is of much interest. Are there results that provide insights into differences between ViT and Swin, for example? The second half of the results section should be modified to include more results and discussion of Swin and DEIT.

(3)	The paper labels very different architectural components with the same nomenclature. Is it really sensible to use “backbone” for both a 50-layer ResNet and a linear patch embedding? The paper claims that this is “for consistency”, but it makes the layer comparison very questionable. A reader thinks that the two architectures are being compared at an equivalent point in their processing, but it is really not the case.

(4)	The AP values for DETR in Table 1 seem very low. Can you explain why?

---

> ### Author Response · Authors · 2026-06-16
>
> We thank the reviewer for their thorough and helpful comments. We have revised several formulations in the manuscript accordingly and further clarify these points below.
>
>
> Requested Changes:
> (1) C3: The shared property across architectures is indeed primarily the gradual change in representations across layers, as evidenced by the experiments in Section 4.7. In contrast to the reviewer’s interpretation, we believe this is a meaningful property because it differs from the behavior typically observed in CNNs and ResNets (see, e.g., Raghu et al. [1]) As we have stated in the Discussion section, the fact that this pattern appears in both DETR and ViT, together with the findings of Raghu et al., suggests that it may be a fundamental property of transformer architectures more generally.
>
> A second shared property we observed is the similarity between ViT and DeiT III in their internal processing, despite their different training strategies; see Section A.1.
>
> [1]: Maithra Raghu, Thomas Unterthiner, Simon Kornblith, Chiyuan Zhang, and Alexey Dosovitskiy. Do Vision Transformers See Like Convolutional Neural Networks? November 2021. URL https://openreview.net/forum?id=Gl8FHfMVTZu
>
> C4: We focused our analysis on the TVMs DETR and ViT because we view them as representing opposite ends of the spectrum of transformer-based vision models. As the reviewer pointed out, the models differ substantially in architecture, objective, and training dataset, and we therefore considered their comparison particularly informative and broadly representative of different aspects of TVMs. Swin and DeiT III were used in addition to establish the feasibility of applying our method to other TVMs. We also provide the analyses for Swin and DeiT~III in the appendix; see Sections A.1 and A.2.
>
> C5: We support this claim not only through our analysis of DETR, but also through the results obtained for Swin. In particular, Table 1 shows that, for images reconstructed from high-level processing stages, Swin achieves substantially higher scores on semantic metrics, including FID, CLIPScore, and accuracy, with the modular approach than with the classic end-to-end approach.
>
> We agree that our original formulation may have suggested that this property holds across all architectures. Our intended claim was more limited: The method can produce more semantically coherent reconstructions from high-level processing stages in architectures with multiple processing stages of interest and significant omission/abstraction of image details, such as DETR and Swin. ViT and DeiT III do not fall into this category. We have therefore refined the wording to make this claim conditional and to avoid overstating its generality
>
> This clarification should also explain the reviewer's observation that, in Figure 5, ViT yields similar reconstructions under the modular and end-to-end approaches, whereas DETR shows a clear difference between the modular and classic approaches.
>
> Regarding the reviewer's statement that the AP values for the modular and fine-tuned DETR variant with lambda = 0 in Figure 4 are equal: The equality is expected because in both variants, the weights of the forward network are not affected by the reconstruction loss; see Section 4.2.3, Paragraph 2. Note that we feed the original input images, not the reconstructed images, into the architecture.
>
> To clarify, the purpose of Figure 4 was not only to demonstrate the capability of modular feature inversion but also to show that the averaging observed in the reconstructed images results from the omission or abstraction of image details in the intermediate representations, and not from insufficiently powerful inverse models. As evidenced by the quantitative and qualitative analyses, image reconstructions can be made more detailed by introducing reconstruction-driven information into the intermediate representation.

---

> ### Author Response · Authors · 2026-06-16
>
> (2) As mentioned above, beyond the pioneering role of DETR and ViT in this domain, we chose to focus on these two architectures because they represent opposite ends of the spectrum of transformer-based vision models. Accordingly, the primary goal of our work was not to reveal differences between DETR and ViT per se, but rather to study TVMs through a comparison of two distinct and representative architectures.
>
> We used Swin and DeiT III primarily to validate our method. However, we also include parts of the corresponding results, albeit in less detail, in the appendix of our manuscript; see Sections A.1 and A.2. Regarding the comparison between ViT and Swin, our method indicates that Swin behaves similarly to ViT. In the final two stages, particularly the last, Swin omits more image details than ViT
>
> Regarding the reviewer’s concern that differences in processing between the two architectures are unsurprising because the architectures were designed with different objectives, our results suggest otherwise: The observed differences are driven primarily by differences in the backbones rather than by the objectives themselves; see Section 4.8.
>
>
> (3) We use the term backbone to refer to the processing that occurs before the transformer stage. Our aim is not to equate a ResNet backbone with a linear patch embedding, but rather to establish a clear distinction between transformer-stage processing and the processing that precedes it. Accordingly, our focus is on examining the role of the backbone in subsequent processing stages within the transformer, rather than studying differences in representations at the backbone stage in isolation. As analyzed in Section 4.8, this preprocessing plays a pivotal role in how the transformer components of the architecture process images. We agree, however, that this terminology may be confusing for readers. To clarify this point, we have emphasized the differences between the backbones in the Methods section of the manuscript.
>
>
> (4) The AP values for DETR, as well as the accuracy values for the other architectures reported in Table 1, are computed on reconstructed images from different processing stages. Specifically, a reconstructed image from a given processing stage is fed back into the forward architecture, and AP or accuracy is then computed; see also the caption of Table 1.
>
> Because DETR omits or abstracts away image details already at the backbone stage, the reconstructed images differ substantially from the input domain across all stages, likely making objects harder to detect and leading to low AP values. This effect is not observed for ViT and DeiT III, which preserve more image details throughout processing. For Swin, a similar but less pronounced effect can be observed at the final processing stage, where the architecture has likewise discarded many image details.

---

> > ### Comment · Reviewer_YFXv · 2026-06-22
> > **Changes to the paper?**
> >
> > I appreciate the author's detailed response, but some of the core criticisms and requests for changes will remain unaddressed if there are no modifications to the paper (I don't currently see a revision).
> >
> > For example, even if the gradual change in representations is viewed as an important shared property, the results/discussion section must at least detail and discuss the shared properties if this remains as a core claim in the introduction. A paragraph or two would suffice. Currently there is barely any discussion and no detail so "We identify shared properties across DETR, ViT, Swin, and DeiT III..." is really not supported by the paper. Pointing to an additional result in the appendix is not sufficient - the main body of the paper should provide clear support for core bulleted claims in the introduction.
> >
> > Other responses like "Our intended claim was more limited:" suggest that a modification of the paper is required.
> >
> > None of the criticisms in the initial review require dramatic changes to the paper or new results, but there is a need for adjustment of claims and/or minor restructuring of the paper.

---

> > > ### Author Response · Authors · 2026-06-23
> > > **Revised manuscript**
> > >
> > > We thank the reviewer for their quick response. We have now uploaded a further revised version of the manuscript. For convenience, all changes are highlighted in blue again.
> > >
> > > Following the reviewer's suggestions, we have emphasized the focus of our study on DETR and ViT, explicitly described the gradual refinement of representations across layers as a shared characteristic of these two architectures rather than referring to shared characteristics, and clarified our use of the term "backbone." We had already addressed the generalization claim in the previous revision in response to Reviewers Huua and Yseu.
> > >
> > > Please let us know if you are unable to access or view the revised manuscript.

---

### Author Response · Authors · 2026-06-04
**Updated Manuscript**

Dear reviewers,

Thank you for your thorough and constructive feedback. We uploaded a new version of the manuscript incorporating the requested changes. For convenience, all changes are shown in blue font.

---

### Author Response · Authors · 2026-06-23
**Revised Manuscript**

Dear Reviewers,

We have uploaded a slightly revised version of our manuscript in response to Reviewer YXFv’s comments. Please refer to our most recent response to Reviewer YXFv for further details.